# Precipitation-Driven Soil and Vegetation Changes Shape Wetland Greenhouse Gas Emissions

**DOI:** 10.3390/biology14121663

**Published:** 2025-11-24

**Authors:** Ziwei Yang, Kelong Chen, Hairui Zhao, Ni Zhang, Desheng Qi

**Affiliations:** 1School of Geographical Sciences, Qinghai Normal University, Xining 810008, China; 15756789182@163.com (Z.Y.); 18697237052@163.com (H.Z.); zhangni0224@163.com (N.Z.);; 2Key Laboratory of Natural Geography and Environmental Processes of Qinghai Province, Xining 810008, China; 3Key Laboratory of Qinghai-Tibet Plateau Surface Process and Ecological Conservation, Ministry of Education, Qinghai Normal University, Xining 810008, China

**Keywords:** precipitation change, greenhouse gas fluxes, soil–vegetation interactions, alpine wetland ecosystem

## Abstract

Climate change is altering rainfall patterns, with more frequent droughts and heavy rains. This study investigates how these changes affect the emission of greenhouse gases from a high mountain wetland. We found that a moderate increase in rain boosted plant growth and carbon dioxide (CO_2_) emissions but reduced methane (CH_4_) and nitrous oxide (N_2_O) emissions. However, both extreme rainfall increase and drought had negative effects: they disrupted the balance between the soil and the gases, leading to lower CO_2_ emissions but more unpredictable CH_4_ and N_2_O releases, while also stunting plant growth. In short, moderate wetting can benefit these fragile ecosystems, but more extreme weather events threaten their health and stability. Our research helps predict how these vital ecosystems will respond to a changing climate.

## 1. Introduction

Since the Industrial Revolution, the global climate system has been undergoing accelerated warming due to the continuous rise in atmospheric concentrations of greenhouse gases (GHGs) such as CO_2_, CH_4_, and N_2_O. According to Intergovernmental Panel on Climate Change (IPCC) assessments, the current atmospheric CO_2_ concentration has increased by approximately 50% relative to pre-industrial levels, while CH_4_ concentrations have risen by more than 160%. Global mean surface temperature has already increased by 1.1 °C and is projected to rise by an additional 1.5–4.0 °C by the end of the 21st century [1]. Although the atmospheric abundance of N_2_O is relatively low, its strong infrared absorption capacity and long atmospheric residence time confer a global warming potential 150–200 times that of CO_2_ [2]. The rapid increase in GHG concentrations is expected to exacerbate global climate change, leading to sea level rise, more frequent extreme climate events, and intensified natural disasters. Against this backdrop, investigating the responses of GHG emissions to changing precipitation patterns is of both scientific and practical significance.

Wetlands, as critical carbon reservoirs within terrestrial ecosystems, occupy only 5–8% of the global land surface but store approximately 500 Pg of soil carbon, accounting for 20–30% of the global soil carbon pool [3,4]. The spatiotemporal variability of wetland GHG emissions is highly dependent on environmental factors such as temperature, water level, and vegetation composition. Methane emissions are primarily regulated by methanogenic activity and oxidation processes, whereas CO_2_ fluxes are closely linked to soil respiration and plant photosynthesis [5]. Global warming accelerates microbial metabolism and organic matter decomposition, thereby directly stimulating GHG emissions, while precipitation changes indirectly alter wetland carbon cycling by modifying hydrological regimes, redox conditions, and vegetation community structure [6]. For instance, elevated soil moisture may suppress CO_2_ release but enhance anaerobic CH_4_ production, whereas drought tends to increase soil aeration, promoting CO_2_ release and inhibiting CH_4_ emissions [7]. Moreover, the anoxic conditions typical of wetlands strongly reshape nitrogen biogeochemical cycling pathways, thereby exerting significant controls on N_2_O fluxes. Owing to their unique hydrological regimes and ecological diversity, wetland responses to environmental change are often complex and pronounced. However, climate-induced warming and altered precipitation regimes are profoundly affecting wetland hydrology and ecosystem processes, potentially shifting their role from carbon sinks to carbon sources [8].

Although precipitation manipulation experiments have been widely conducted in grassland and lowland wetland ecosystems, studies focusing on alpine headwater wetlands remain limited, particularly regarding their long-term responses to altered precipitation regimes and vegetation–soil interactions at the plot scale. The long-term hydrological sensitivity and complex feedbacks between soil moisture, microbial activity, and plant community dynamics in these systems are still poorly understood.

The Qinghai Lake Basin, located on the northeastern margin of the Qinghai–Tibet Plateau, represents a typical endorheic inland watershed, and its wetland ecosystems exhibit regionally representative responses to climate change. In recent decades, mean annual air temperature in this basin has increased at a rate of 0.3–0.4 °C per decade, while precipitation has shown a spatially heterogeneous pattern characterized by “increasing in the south and decreasing in the north” [9]. Hydrological observations indicate that the water level of Qinghai Lake rose by 1.5 m between 2004 and 2020; nevertheless, extreme drought events (e.g., the summer of 2022) still caused seasonal desiccation of local wetlands [10]. This dynamic imbalance between precipitation and evapotranspiration has markedly altered flooding cycles and soil moisture regimes, with cascading effects on vegetation distribution and microbial metabolism.

Compared with previous precipitation manipulation studies in alpine or plateau regions, the present experiment extends over multiple years and employs a wider precipitation gradient (+75%, +25%, −25%, and −75%) with in situ replicated plots, enabling the assessment of nonlinear responses and ecosystem resilience under extreme conditions.

Furthermore, we hypothesize that changes in precipitation modify soil water content and redox potential, thereby affecting methanogenesis, denitrification, and soil respiration, while vegetation responses (e.g., biomass and root activity) further regulate substrate supply and CO_2_ flux. This conceptual framework links hydrological variation to microbial processes and plant-mediated carbon cycling, forming the mechanistic basis of this study.

In this study, we focus on alpine wetlands within the Qinghai Lake Basin to investigate how precipitation changes indirectly influence CO_2_, CH_4_, and N_2_O fluxes by regulating soil physicochemical properties (e.g., moisture content, salinity) and vegetation community attributes (e.g., biomass, functional types). Specifically, we address the following scientific questions: (1) How does precipitation reduction alter soil moisture dynamics and, in turn, microbial activity and carbon decomposition? (2) How do vegetation adaptive strategies mediate the impacts of precipitation change on GHG emissions? (3) What are the feedback effects of soil–vegetation interactions on regional carbon cycling? The findings of this study are expected to provide a theoretical basis for the adaptive management of wetlands on the Qinghai–Tibet Plateau and to supply essential data for improving carbon–climate models of alpine ecosystems at the global scale.

## 2. Materials and Methods

### 2.1. Study Area

Qinghai Lake Basin was selected because it is one of the most climate-sensitive alpine wetland systems on the northeastern Qinghai–Tibet Plateau, characterized by Kobresia-dominated vegetation, strong hydrological fluctuations, and high sensitivity of greenhouse gas emissions to precipitation variability. [11]. Elevation within the basin ranges from 3600 to 5000 m. Field experiments were conducted at the Wayan Mountain Wetland Comprehensive Observation Station in Yike’ulan Township, Gangcha County, Haibei Tibetan Autonomous Prefecture, northern Qinghai Province. The geographic coordinates of the site are 37.73° N and 100.08° E, with an elevation ranging from 3720 to 3850 m. The Wayan Mountain alpine wetland monitoring station lies north of the Qilian Mountains and south of Qinghai Lake, approximately 52 km northwest of Gangcha County, and represents the headwater wetland of the Wayanqu River, a tributary of the Shaliu River [12].

The region is predominantly influenced by southwesterly winds. Annual cumulative solar radiation reaches 6938.3 MJ·m^−2^, and the mean annual specific humidity is 3.7 g·kg^−1^ [11]. The average annual precipitation is 426.8 mm, over 90% of which occurs during the plant growing season from May to September [13]. The mean annual air temperature is −2.6 °C, with daily mean maxima of 14.4 °C in summer and minima of approximately −23.8 °C in winter. The freeze period lasts for about 145 days [14], classifying the region as a typical alpine continental climate [15]. The soils in the precipitation manipulation plots are primarily Histic Cambisol (soil classification followed the World Reference Base for Soil Resources) with an average depth of 1.2 m, underlain by alluvial deposits [16]. Previous studies have shown that the upper ~1.7 m of soil in the Wayan Mountain alpine marshland represents seasonally frozen soil, while deeper layers are underlain by permafrost. Meteorological records indicate that the top 5 cm of soil begins to freeze in mid-November and thaws in early April, with a freezing period of ~145 days annually [17]. Variations in air temperature correspond closely to changes in the 5 cm soil temperature, peaking in August and reaching a minimum in January [18].

Vegetation within the Wayan Mountain observation area is relatively homogeneous. The dominant species is *Kobresia humilis*, accompanied by *Carex tristachya*, *Lobularia maritima*, and *Potentilla anserina*. Vegetation coverage exceeds 90%, with bare patches accounting for more than 4% of the surface area (Table 1) [19].

### 2.2. Experimental Design and Sample Collection

During the growing seasons (May–September) of 2020 and 2021, field experiments were conducted at the Wayan Mountain Observation Station to simulate greenhouse gas (GHG) emissions from headwater wetlands under different precipitation regimes. Experimental plots were established in August 2018 in an open area with relatively uniform topography and vegetation growth.

The experimental area measured 30 m × 30 m in total, within which nine subplots (3.2 m × 2.6 m each) were embedded. The remaining space within the 30 m × 30 m plot served as buffer zones between subplots (3 m between adjacent plots) and as an outer protective buffer (5 m surrounding the plot). Thus, the overall 30 m × 30 m area encompassed both experimental subplots and required buffer space.

The nine subplots were arranged in a 3 × 3 grid. Each column represented a treatment group, while each row contained three replicates of the same treatment. To prevent surface runoff, each subplot was bordered by a 20 cm runoff barrier, constructed by embedding galvanized iron sheets 25 cm below ground. Within each subplot, a 40 cm × 40 cm static chamber base was installed in the northwest corner for GHG sampling, a 1 m × 1 m permanent quadrat was set in the northeast for vegetation monitoring, and the southern section was designated for soil and random vegetation sampling.

This experiment followed a single-factor randomized block design, consisting of five precipitation treatments with three replicates each. Precipitation was the primary experimental variable, with one control and four manipulated treatments: natural precipitation (CK), +25% rainfall, −25% rainfall, +75% rainfall, and −75% rainfall. Rainfall reduction treatments were achieved using concave, transparent rain-sheltering panels that intercepted 25% or 75% of precipitation. These panels were installed at an 11° inclination to facilitate water collection via gravity. The intercepted water was stored in side-mounted troughs and subsequently directed into PVC pipes (50 cm above ground, spaced 40 cm apart) to redistribute the collected water evenly into the +25% and +75% rainfall plots, thereby simulating increased precipitation.

#### 2.2.1. Precipitation Simulation Device

The rainfall manipulation shelters consisted of upward-facing U-shaped panels that intercepted either 25% or 75% of the plot area. Each shelter was installed with an 11° slope to facilitate runoff. Collected rainwater was directed through U-shaped gutters into PVC pipes, which were spaced 40 cm apart and perforated with sprinkling holes. These pipes redistributed the captured water evenly across the designated +25% and +75% rainfall plots, while the control plots (CK) remained unaltered. No rainfall manipulation shelters were installed in the control (CK) plots.

An automated water diversion system (Figure 1) was employed to simulate precipitation. The system was composed of a rainwater guide trough, a confluence channel, a storage barrel, and a sprinkling unit, with each component arranged at progressively lower heights. This gravity-driven design allowed rainwater to be collected and redistributed automatically without the need for additional energy inputs such as electricity or fuel [20].

The 25% and 75% rainfall reduction treatments were achieved by adjusting the horizontal coverage area of the U-shaped panels to intercept exactly 25% or 75% of the plot surface. Intercepted water was routed into gutters and redistributed evenly through perforated PVC pipes.

#### 2.2.2. Experimental Design

The mean annual precipitation at the Wayan Mountain station is approximately 426.8 mm. According to data from the Gangcha County Hydrological Bureau, regional precipitation has generally remained above average since 2005. Based on this baseline, the simulated precipitation treatments were designed as follows: +25% (≈533.5 mm), −25% (≈320.1 mm), +75% (≈746.9 mm), and −75% (≈106.7 mm). The purpose of these treatments was to mimic the impacts of extreme increases and decreases in precipitation on vegetation, soil properties, and microbial communities, and thereby assess their effects on ecosystem greenhouse gas (GHG) emissions.

The precipitation manipulation system was established in 2018 and has operated continuously for four years without human intervention or external energy input. Due to seasonal freeze–thaw processes and burrowing activities of plateau pikas, the local topography is uneven; therefore, relatively flat terrain within the experimental site was selected for monitoring. The experimental plots were enclosed within a 30 × 30 m area and subdivided into nine subplots (3.2 × 2.6 m each), arranged in a 3 × 3 grid. Each treatment was replicated three times. Subplots were separated by 3 m buffer strips, and the entire experimental plot was surrounded by a 5 m buffer zone. To prevent surface runoff, each subplot was bordered by a 20 cm runoff barrier, created by embedding galvanized iron sheets 25 cm underground. Within each subplot, a 40 × 40 cm static chamber base was installed in the northwest corner for GHG sampling, a 1 × 1 m permanent quadrat was placed in the northeast for vegetation monitoring, and the southern section was designated for soil and random vegetation sampling.

The experimental treatments included +25% rainfall, −25% rainfall, +75% rainfall, −75% rainfall, and a natural control. With increasing soil depth, the temporal variation in soil temperature became less pronounced, showing only a seasonal pattern (Figure 2). Compared with 2019, the 2020 growing season featured greater early-season precipitation, while 2021 was relatively cooler and drier. Such interannual differences help explain the lower CO_2_ fluxes observed in 2021.

#### 2.2.3. Gas Sampling and Analysis

Greenhouse gas (GHG) fluxes of CO_2_, CH_4_, and N_2_O were measured using the static chamber–gas chromatography method (Agilent Technologies, Inc., 2850 Centerville Road Wilmington, DE, USA) [21].

Each static chamber measured 40 cm × 40 cm in length and width, with a height of 20 cm. The stainless-steel base was installed flush with the soil surface and remained in place throughout the experimental period. Before gas collection, the groove surrounding each chamber base was filled with water approximately 10 min in advance to ensure an airtight seal.

Gas sampling was conducted at 0, 15, and 30 min after chamber closure using 100 mL gas-tight syringes fitted with three-way stopcocks (Kangfulai Medical Supplies Co., Ltd., Danyang, China). The temperature inside the chamber was recorded during each sampling to correct for temperature-induced concentration variations. Sampling was performed twice daily, at 11:00 and 13:00, on clear days during early and late periods of each month from May to September in 2020 and 2021. These sampling times were selected because previous studies in similar alpine wetlands have identified 11:00–13:00 as representative of the diurnal mean flux period.

Gas samples were analyzed using an Agilent 7890B gas chromatograph equipped with a thermal conductivity detector (TCD) for CO_2_, a flame ionization detector (FID) for CH_4_, and a ^63^Ni electron capture detector (ECD) for N_2_O (Agilent Technologies, Inc., Wilmington, DE, USA). Calibration was performed prior to each monthly measurement using two standard gases, with concentrations of CO_2_ = 606.6 × 10^−6^ ppm, CH_4_ = 10.1 × 10^−6^ ppm, and N_2_O = 1.0 × 10^−6^ ppm. Detection limits for CO_2_, CH_4_, and N_2_O were consistent with the manufacturer’s specifications for the Agilent 7890B system.

Fluxes were calculated using the standard formula for wetland greenhouse gas emissions [22]. When concentration–time relationships were nonlinear, the initial 30 min linear segment was used to estimate fluxes. Blank control chambers were installed in undisturbed, natural wetland plots to represent baseline fluxes without precipitation manipulation.

Flux calculation equation [22]:(1)$$F=\rho \times \frac{V}{A}\times \frac{P}{P_{0}}\times \frac{T_{0}}{T}\times \frac{dC_{t}}{d_{t}}$$
where F is the gas flux (mg·m^−2^·h^−1^), ρ is the gas density (mg·mL^−^1), V is the chamber volume (m^3^), A is the chamber base area (m^2^), dCt/dt is the rate of concentration change over time (mg·m^−3^·h^−1^), and T is the mean air temperature inside the chamber (°C), P represents the standard atmospheric pressure, P_0_ represents the atmospheric pressure at the sampling site, and T_0_ denotes the temperature inside the chamber at the time of sampling.

Estimation of Total Greenhouse Gas Emissions and Global Warming Potential.

The cumulative emissions of greenhouse gases were estimated using the following equation [23]:(2)$$M=\sum (F_{i+1}+F_{i})/2\times (t_{i+1}-t_{i})\times 24$$
where M is the cumulative flux of the target gas (kg·hm^−2^); F_i_ and F_i+1_ represent the gas fluxes at the ith and (i+1)th sampling events (µg·m^−2^·h^−1^ or mg·m^−2^·h^−1^); t_i_ and t_i+1_ are the sampling dates of the ith and (i+1)th events, respectively.

Global warming potential (GWP) was used to convert CH_4_ and N_2_O fluxes into CO_2_-equivalent emissions. GWP expresses the cumulative radiative forcing of a greenhouse gas relative to carbon dioxide (CO_2_) over a defined time horizon. As established by the Intergovernmental Panel on Climate Change (IPCC), this metric integrates both the radiative efficiency of a gas and its atmospheric lifetime. Under the 100-year horizon, the GWP of CH_4_ and N_2_O is 28 and 298 relative to CO_2_, respectively. Thus, Equation (3) quantifies the overall warming effect of the three gases emitted from the wetland by expressing them in unified CO_2_-equivalent terms [23]:(3)$$GWP=F_{CO_{2}}+F_{CH_{4}}\times 25+F_{N_{2}O}\times 298$$
where GWP represents the global warming potential (t·hm^−2^); FCO_2_, FCH_4_, and FN_2_O denote the cumulative emissions of CO_2_, CH_4_, and N_2_O (t·hm^−2^) during the observation period; and the constants 25 and 298 represent the 100-year time-horizon GWP factors of CH_4_ and N_2_O relative to CO_2_, respectively.

In this study, no direct measurements of surface or groundwater table depth were conducted. Instead, soil moisture at 10 cm depth was used as a proxy to represent near-surface hydrological conditions. We acknowledge this as a limitation and plan to include water table monitoring and microbial process indicators in future research.

#### 2.2.4. Soil and Vegetation Sampling

During the sampling period, soil temperature at 10 cm depth was measured using a soil thermometer (TZS-2X, precision ±0.01 °C) (Top Cloud-agri China, Hangzhou, China), and soil moisture was measured using a soil moisture meter (JK-100F, precision ±0.1%) (Shanghai Tuoxi Electronics Technology Co., Ltd., Shanghai, China). Data were subsequently analyzed to characterize their temporal dynamics. Soil samples were randomly collected from three locations within each quadrat at depths of 0–10 cm and 10–20 cm. Total carbon (TC) and total nitrogen (TN) were determined using an elemental analyzer (ECS4024, LICA United Technology Limited, Beijing, China); soil pH was measured with a pH meter (PHS-25) (Shanghai Instrument and Electrical Scientific Instruments Co., Ltd., Shanghai, China); and soil electrical conductivity (EC) was determined using a conductivity meter (DDS-307) (Shanghai Instrument and Electrical Scientific Instruments Co., Ltd., Shanghai, China).

Figure 2 illustrates the seasonal progression of daily precipitation and 10 cm soil temperature from December 2019 to December 2021. Soil temperature followed a clear annual cycle, rising steadily from early spring, peaking between July and August, and declining sharply from late autumn into winter. Precipitation was concentrated in the growing season (May–September), forming distinct rainfall peaks during mid-summer, while winter precipitation remained extremely low. These joint temporal patterns reflect the typical alpine continental climate of the Qinghai–Tibet Plateau and provide essential environmental context for interpreting the greenhouse gas fluxes observed in 2020 and 2021.

Aboveground biomass was measured monthly from May to September in 2020 and 2021 using the clipping method. Specifically, in each treatment and control group, three replicate plots were established, and vegetation was sampled randomly within 25 × 25 cm quadrats. All aboveground plants were clipped, and litter was removed before placing samples into sealed bags for laboratory analysis. Plant materials were first inactivated at 105 °C for 30 min, followed by oven-drying at 65 °C for 24 h, and then weighed. Biomass values were standardized to unit area.

Belowground biomass was determined by root coring after clipping. Soil samples (0–10 cm) were collected using a 4.2 cm diameter root auger within the clipped quadrats. In the laboratory, soils were passed through a 2 mm sieve to separate root material. Roots were washed free of soil, oven-dried at 85 °C for 24 h, and then weighed.

For TC and TN determination, soil samples from 0 to 10 cm and 10 to 20 cm were air-dried, sieved through a 2 mm mesh, homogenized, and further ground to pass through a 0.25 mm sieve. Approximately 20 mg of soil was weighed using a high-precision balance (10^−6^ g) and combusted at high temperature with the elemental analyzer (ECS4024, LICA United Technology Limited, Beijing, China).

Soil pH was measured by mixing 10 g of air-dried soil (<2 mm) with 25 mL of ultrapure water, shaking thoroughly, and allowing to settle for 30 min before measurement using a pH meter (PHS-25) (Shanghai Instrument and Electrical Scientific Instruments Co., Ltd., Shanghai, China). Soil EC was determined by mixing 10 g of sieved soil with 50 mL of ultrapure water, shaking thoroughly, filtering, and measuring the filtrate with a conductivity meter (DDS-307) (Shanghai Instrument and Electrical Scientific Instruments Co., Ltd., Shanghai, China).

Vegetation cover (%) was visually estimated within each 1 × 1 m quadrat using standard cover-class estimation methods.

Vegetation characteristics were calculated using the following formulas:(4)$$P_{i}=(C+D+H)/3$$
where P_i_ is the relative proportion of individuals of species i to the total number of individuals; C, D, and H represent relative coverage, relative density, and relative height, respectively.

Biodiversity indices were calculated as follows:(5)$$\mathrm{Simpson}’s\;\mathrm{Index}\;(D):\;C=1-\sum_{i}^{S}P_{i}^{2}$$(6)$$\mathrm{Shannon}–\mathrm{Wiener}\;\mathrm{Diversity}\;\mathrm{Index}\;(H^{\prime }):\;H^{\prime }=-\sum P_{i}InP_{i}$$(7)$$\mathrm{Pielou}’s\;\mathrm{Evenness}\;\mathrm{Index}\;(J):\;J=H^{\prime }/InS$$
where S is the total number of species recorded in each quadrat.

### 2.3. Data Analysis

All data visualization was performed using OriginPro 2021, while statistical analyses were conducted in SPSS 21. Based on the significant variations in environmental variables observed under different precipitation treatments, one-way analysis of variance (ANOVA) with least significant difference (LSD) post hoc tests was applied to evaluate the responses of precipitation level, aboveground and belowground biomass, soil moisture (10 cm), soil temperature (10 cm), soil total nitrogen (TN), soil total carbon (TC), soil pH, soil electrical conductivity (EC), and greenhouse gas (GHG) fluxes to precipitation manipulation. OriginPro 2021 was further used to process the temporal dynamics and trends of GHG fluxes and associated environmental variables under different treatments and to generate the corresponding figures. Pearson correlation coefficients (r) were calculated using n = 30 observations per variable pair (biweekly measurements × 5 treatments × 3 replicates).

## 3. Results

### 3.1. Responses of Wetland Greenhouse Gases to Precipitation Manipulation

During two consecutive growing seasons, CO_2_ fluxes in the headwater wetlands exhibited pronounced seasonal dynamics under different precipitation treatments, generally following a “rise-then-decline” pattern. In 2021, the seasonal fluctuations were relatively smoother compared with 2020, likely because long-term precipitation manipulation reduced variability among subplots. Seasonal mean values remained consistently positive across all treatments, indicating that the wetland functioned as a net CO_2_ source (Figure 3).

In 2020, the ranking of CO_2_ fluxes among treatments was as follows: +25% > CK > −75% > +75% > −25%. Notably, under the +25% treatment, the mean growing-season CO_2_ flux reached 101.45 mg·m^−2^·h^−1^, which was significantly higher than that of the control and other treatments (*p* < 0.05, one-way ANOVA with LSD post hoc test). In 2021, the treatment response pattern was generally consistent with that of 2020, with the +25% precipitation treatment still exerting the strongest stimulation on CO_2_ emissions. However, the magnitude of fluxes decreased by nearly half, with a seasonal mean of 51.85 mg·m^−2^·h^−1^ (Figure 3). This interannual reduction may be attributed to lower air temperature and precipitation in 2021 compared with 2020, resulting in reduced microbial respiration and vegetation photosynthetic activity (see Table 1).

Effect sizes (η^2^) ranged from 0.34 to 0.46 across treatments, indicating a moderate-to-strong influence of precipitation on CO_2_ flux. Standard deviations (rather than standard errors) were calculated to better capture plot-level variability. Sampling frequency and chamber sealing precision may introduce small uncertainties (<5%) in the estimated fluxes, but these do not alter the observed treatment patterns.

### 3.2. Responses of CH_4_ Fluxes to Precipitation Manipulation

Seasonal variation in CH_4_ fluxes was not pronounced. In the 2020 growing season, only the +75% precipitation treatment exhibited a weak source with a positive flux (0.56 µg·m^−2^·h^−1^), while all other treatments, including the control, functioned as CH_4_ sinks. The ranking of treatments was: +75% > CK > +25% > −75% > −25% (Figure 4). In 2021, all treatments consistently acted as CH_4_ sinks, with the order of fluxes as: CK > +25% > +75% > −75% > −25%. Even under extreme precipitation increase, the wetland ecosystem still maintained a net absorption state. Notably, CH_4_ fluxes remained relatively stable during the peak of the growing season—when precipitation and temperature were highest—whereas greater fluctuations were observed in the early and late stages of the season (Figure 4).

The persistence of CH_4_ sink behavior under high-moisture conditions may be explained by strong methanotrophic activity in the oxygenated surface layer and efficient CH_4_ oxidation within plant rhizospheres. Previous studies on alpine wetlands have shown that methanotrophs can remain active even under fluctuating water levels due to high soil porosity and root-mediated gas transport [24].

Notably, CH_4_ fluxes remained relatively stable during the peak of the growing season—when precipitation and temperature were highest—whereas greater fluctuations were observed in the early and late stages of the season (Figure 4). Although the absolute flux values were low, precipitation treatments significantly affected CH_4_ fluxes (*p* < 0.05, ANOVA + LSD), with a mean effect size (η^2^) of 0.29. Standard deviations are displayed in Figure 4 to reflect the range of spatial variation across replicate plots.

### 3.3. Responses of N_2_O Fluxes to Precipitation Manipulation

N_2_O is produced through a variety of biotic and abiotic processes, often accompanied by concurrent reduction or consumption [25], making its emission dynamics in wetlands particularly complex. During the 2020 growing season, the +75% precipitation treatment acted as a source (2.03 µg·m^−2^·h^−1^), while the −75% treatment showed weak emissions (0.92 µg·m^−2^·h^−1^). Both ± 25% treatments functioned as weak sinks. The overall ranking of fluxes was: +75% > −75% > CK > −25% > +25%. In contrast, during the 2021 growing season, the −75% treatment functioned as a source (1.08 µg·m^−2^·h^−1^), whereas all other treatments exhibited sink behavior, with the order: −75% > CK > +75% > +25% > −25% (Figure 5).

Short-term increases in soil moisture appeared to stimulate denitrification, enhancing N_2_O emissions, whereas prolonged saturation limited oxygen diffusion and suppressed nitrification, reducing overall fluxes. Precipitation treatments significantly affected N_2_O emissions (*p* < 0.05, ANOVA), though interannual variability was high (η^2^ = 0.21).

### 3.4. Relationships Between Soil Physicochemical Properties and Greenhouse Gas Fluxes Under Precipitation Manipulation

Greenhouse gas (GHG) emissions in wetland ecosystems result from the interplay of a series of biochemical and physical processes. During the 2020 growing season, soil moisture under the +25% and +75% treatments did not differ significantly from that of the control (CK), while the −25% and −75% treatments reduced soil moisture by 3.37% and 2.57%, respectively. Despite warming and increased precipitation, soil moisture did not increase significantly overall, remaining similar to CK, with reduced declines under precipitation reduction. On average, soil temperature under precipitation reduction was 0.53 °C higher than under increased precipitation, indicating that additional rainfall significantly decreased soil temperature. In 2021, soil moisture increased by 0.60% and 1.89% under the +25% and +75% treatments, respectively, whereas the −25% and −75% treatments reduced moisture by 2.08% and 2.30%.

Overall, CH_4_ fluxes in alpine headwater wetlands were primarily controlled by soil temperature (*p* < 0.05), generally showing a negative correlation with soil moisture. Soil pH influenced the activity of methanogenic communities, thereby affecting CH_4_ production and oxidation. Under the +25% treatment, soil pH was about 5.52 and showed a significant negative correlation with CH_4_ flux (*p* < 0.05), while under the −75% treatment, pH was 5.26 and showed a significant positive correlation. Soil pH strongly shapes dominant microbial communities: when pH > 7, acetoclastic methanogenesis is dominant, whereas at lower pH values, the H_2_/CO_2_ pathway prevails. CH_4_ fluxes showed no significant correlation with soil TN or TC across treatments, consistent with findings by Geng et al. [25] in grassland ecosystems.

For CO_2_ fluxes, the +25% treatment significantly enhanced emissions, approximately doubling fluxes relative to the other treatments, and showed a strong positive correlation with soil TC. Soil temperature was generally higher in precipitation reduction plots compared with precipitation addition plots, which would normally enhance microbial activity and respiration. However, CO_2_ fluxes were not significantly correlated with soil temperature (*p* > 0.05), suggesting that soil moisture, rather than temperature, was the dominant factor controlling CO_2_ emissions in headwater wetlands (Table 2 and Table 3).

Across both growing seasons, N_2_O fluxes were not significantly correlated with soil temperature or moisture. Typically, rising temperatures increase soil respiration, reducing oxygen availability and promoting denitrification. In this study, N_2_O fluxes showed a significant negative correlation with soil pH (*p* < 0.05). Nitrification is highly sensitive to soil pH, and precipitation reduction produced more acidic conditions compared with precipitation addition. However, the overall soil pH of the Wayan Mountain headwater wetlands was not favorable for N_2_O emissions, as the optimal range for nitrification is 7.5–8.2.

### 3.5. Responses of Vegetation to Precipitation Manipulation

Observations across two consecutive growing seasons revealed that moderate increases in precipitation enhanced vegetation abundance, while excessive water input suppressed growth and diversity. The +25% precipitation treatment significantly increased aboveground biomass and species richness (*p* < 0.05, ANOVA + LSD), whereas the +75% treatment led to waterlogging, reducing root aeration and diversity indices.

During growth, plants absorb soil carbon and nitrogen while competing with soil microorganisms for nutrients [26]. Among all treatments, +25% precipitation exerted the most pronounced positive effect on plant growth and contributed to greater community diversity (Table 4).

## 4. Discussion

### 4.1. Effects of Increased Precipitation on Wetland Greenhouse Gas Emissions

One of the defining features of wetlands is abundant water, yet changes in precipitation still exert substantial impacts on wetland ecosystems [27]. Altered precipitation modifies soil moisture and subsequently affects vegetation, soil biogeochemistry, and microbial processes, collectively regulating greenhouse gas (GHG) fluxes. Under long-term precipitation manipulation, variation in vegetation traits and soil properties among subplots contributed to heterogeneous GHG responses.

During the 2020 and 2021 growing seasons, CO_2_ fluxes were consistently influenced by precipitation additions. Under the +25% treatment, CO_2_ fluxes increased by 2.35% and 15.94% relative to the control, whereas the +75% treatment decreased fluxes by 37.16% and 43.76%, respectively. Similarly, CH_4_ and N_2_O fluxes exhibited contrasting responses depending on precipitation level and year.

Moderate precipitation addition (+25%) likely enhanced CO_2_ emissions by stimulating vegetation growth and increasing root and microbial respiration through improved water availability. This interpretation aligns with previous studies suggesting that modest increases in soil moisture enhance substrate diffusion and photosynthetic activity while maintaining aerobic microsites necessary for efficient respiration [28].

In contrast, excessive water input (+75%) reduced CO_2_ fluxes, likely due to pore-space saturation, inhibited oxygen diffusion, and reduced microbial respiration—mechanisms widely reported in saturated wetland systems [29,30,31].

For CH_4_, the inhibitory effect of +25% precipitation and the inconsistent response under +75% precipitation suggest that methanogenesis was strongly constrained by redox dynamics and competition with methanotrophy. Increased soil moisture may have reduced surface soil aerobic zones, but strong methanotrophic activity likely counteracted methanogenic production, resulting in persistent CH_4_ sink behavior even under high moisture [32,33].

N_2_O responses also reflected moisture-driven shifts between nitrification and denitrification. Short-term increases in moisture (e.g., +75% in 2020) may have stimulated denitrification, enhancing N_2_O release, whereas prolonged saturation in 2021 likely inhibited nitrification and facilitated complete denitrification *to* N_2_, reducing N_2_O fluxes.

### 4.2. Effects of Reduced Precipitation on Wetland Greenhouse Gas Emissions

Reduced precipitation directly modifies soil physicochemical conditions by enhancing aeration, shifting temperature gradients, and altering nutrient availability. These changes affect microbial activity, organic matter decomposition, and vegetation functioning [34]. Our results demonstrated that both CO_2_ and CH_4_ fluxes declined markedly under −25% and −75% precipitation treatments.

Declines in CO_2_ flux under precipitation reduction likely reflect diminished plant photosynthesis and respiration due *to reduced stomatal conductance*, lower *leaf area*, and decreased water availability. Reduced *soil* moisture also limits *substrate* diffusion to microbial communities, resulting in *weakened* heterotrophic *respiration* [35,36].

CH_4_ fluxes showed substantial reductions under both low-precipitation treatments, suggesting strong constraints on methanogenic production due to enhanced soil aeration and reduced anaerobic microsites.

N_2_O responses diverged between −25% and −75% treatments. Moderate drought (−25%) suppressed N_2_O emissions by limiting nitrification substrates and reducing microbial activity, whereas severe drought (−75%) increased N_2_O release, potentially due to increased nitrifier-denitrification or enhanced re-oxidation of nitrification intermediates under fluctuating redox conditions.

These contrasting patterns indicate that drought severity determines nitrogen cycling pathways and associated gaseous losses.

### 4.3. Effects of Soil Properties on Greenhouse Gas Emissions

Soil moisture and temperature are critical regulators of wetland respiration and GHG exchange [37,38]. Altered precipitation reshapes hydrological and thermal conditions, thereby influencing vegetation productivity and microbial metabolism. In this *study*, *CO_2_ fluxes were positively correlated with soil* total *carbon* (TC), *suggesting that* substrate supply, rather than temperature, was the primary factor supporting CO_2_ production. Moderate precipitation increases *may* improve substrate diffusion *and root* turnover, *enhancing* heterotrophic respiration. For CH_4_, *negative correlations with soil C and N factors*—*particularly under precipitation addition*—reflect the dominance of methane oxidation in the oxygenated upper soil layers. The contrasting pH–CH_4_ relationships under +25% (negative correlation) and −75% (positive correlation) treatments support shifts between methanogenic pathways. At lower pH, the H_2_/CO_2_ pathway dominates methanogenesis, whereas under less acidic conditions, acetoclastic *methanogenesis* may become *more* active [39]. For N_2_O, correlations with TN (**Sampling Year: 2020**) *and* pH/*EC* (**Sampling Year: 2021**) suggest that *precipitation*-driven *changes in nitrogen forms and ion distributions* modulate nitrification–denitrification *processes.*

Critically, correlations between soil properties and GHG fluxes weakened under extreme precipitation levels (±75%), indicating nonlinear responses. This weakening suggests that under extreme drought or waterlogging, GHG emissions are shaped by more complex interactions involving vegetation stress, microbial community shifts, and physical constraints on gas–water diffusion, rather than simple linear relationships with soil variables [40,41].

### 4.4. Effects of Precipitation Changes on Wetland Vegetation

Precipitation gradients induced significant differences in vegetation growth, likely reflecting heterogeneity in surface soil moisture across subplots. In headwater wetlands dominated by Kobresia humilis, whose roots are concentrated in the top 0–20 cm soil, increased water availability enhanced plant growth through efficient water uptake. Moderate precipitation addition (+25%) consistently increased plant height, cover, and above- and belowground biomass, likely by alleviating moisture limitations and improving nutrient mineralization. In contrast, severe drought (−75%) significantly reduced biomass and height, demonstrating the strong sensitivity of alpine wetland vegetation to water stress. Excessive precipitation (+75%) produced more variable responses. Although biomass increased in some cases, community diversity indices (Shannon, Pielou) did not correspondingly improve.

This suggests that excess water altered competitive hierarchies, favoring fast-growing or tolerant species while suppressing others, resulting in fluctuating diversity patterns despite biomass increases.

Overall, moderate precipitation addition supported vegetation productivity and diversity, while extreme precipitation—either too high or too low—disrupted community structure and stability.

## 5. Conclusions

This study demonstrates that precipitation changes significantly influence greenhouse gas (GHG) emissions and environmental characteristics of the Qinghai Lake littoral wetland ecosystem by regulating soil moisture conditions. Moderate precipitation addition (+25%) enhanced CO_2_ emissions, vegetation growth, and community productivity, while simultaneously reducing CH_4_ and N_2_O fluxes, highlighting the positive role of moderate water supplementation in sustaining carbon cycling and vegetation functions. In contrast, extreme precipitation alterations (+75% and −75%) weakened the correlations between GHG fluxes and soil factors, resulting in reduced CO_2_ fluxes, greater fluctuations in CH_4_ and N_2_O emissions, and suppressed vegetation growth and community diversity, underscoring the high sensitivity of wetlands to extreme hydrological stress.

The dominant controls on different GHGs varied: CO_2_ emissions were more strongly correlated with soil carbon pools, CH_4_ fluxes were closely associated with soil moisture, and N_2_O dynamics were influenced by the combined effects of soil nitrogen and pH. These relationships reflect observations from a single alpine wetland over two growing seasons, and thus the findings should be interpreted within this spatial and temporal context.

Overall, moderate increases in water availability may help maintain carbon cycling processes and vegetation diversity in alpine wetlands, whereas extreme drought or waterlogging can reduce ecosystem stability. These results suggest that altered precipitation regimes under future climate change could substantially affect GHG dynamics and vegetation structure in alpine littoral wetlands, with implications for regional carbon cycling and ecosystem functioning.

Future research should integrate continuous water table monitoring, microbial functional assays (e.g., methanogenesis and denitrification potentials), and porewater nitrogen assessments to better constrain the mechanistic pathways underlying GHG responses. Long-term observations and multi-site comparisons across the Qinghai–Tibet Plateau will also be essential for improving model predictions of alpine wetland feedbacks to climate change.

## Figures and Tables

**Figure 1 biology-14-01663-f001:**
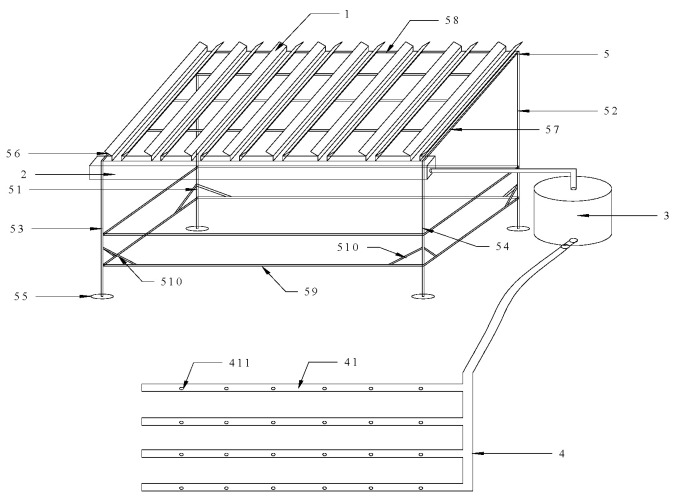
Precipitation Simulator. In the drawing: 1. Rainwater diversion groove, 2. Convergence groove, 3. Sleeping box, 4. Spraying device, 41. Spray pipe, 411. Spray hole, 5. Support frame, 51. First longitudinal rod, 52. Second longitudinal rod, 53. Third longitudinal rod, 54. Fourth longitudinal rod 55. Fixed chassis, 56. First steel pipe, 57. Second steel pipe, 58. Adapting pipe. 59. Cross bar, 510. Oblique strut.

**Figure 2 biology-14-01663-f002:**
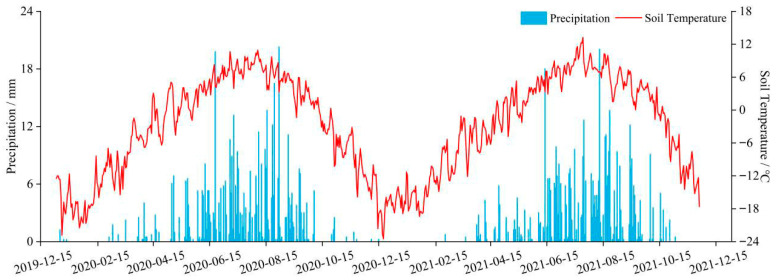
Soil temperature at 10 cm depth and precipitation in the Wayan Mountain plots during the 2020 and 2021 growing seasons.

**Figure 3 biology-14-01663-f003:**
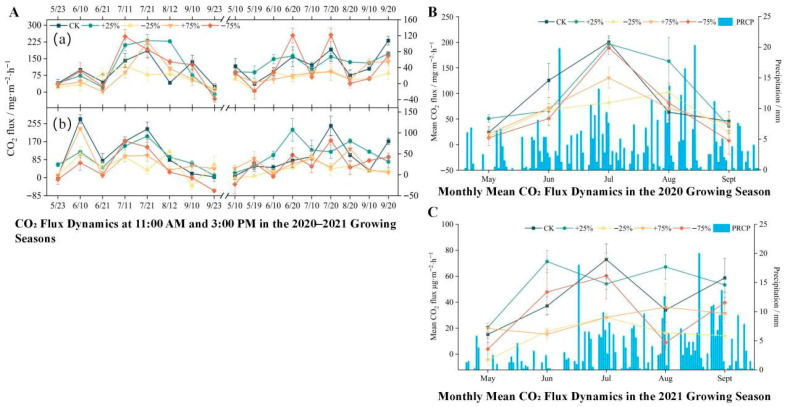
(**A**) Temporal and seasonal variation in ecosystem CO_2_ flux under different precipitation treatments during the 2020–2021 growing seasons. Diurnal CO_2_ flux dynamics at 11:00 a.m. (**a**) and 3:00 p.m. (**b**) from May to September in 2020 and 2021 under five precipitation treatments: control (CK), +25%, −25%, +75%, and −75%. Each point represents the mean flux of a specific sampling date, with error bars indicating the standard error of the mean (n = 3). (**B**) Monthly mean CO_2_ flux (line plots, left *Y*-axis) and daily precipitation (blue bars, right *Y*-axis) during the 2020 growing season. The flux values represent the average of all biweekly measurements within each month under each precipitation treatment. Error bars represent standard error (n = 6 per month). (**C**) Monthly mean CO_2_ flux and daily precipitation during the 2021 growing season, following the same structure as in panel B. The precipitation values in panels B and C are shared across all treatments as background climate conditions, while CO_2_ flux differs by treatment. All CO_2_ fluxes are expressed in mg·m^−2^·h^−1^. Precipitation is measured in mm. The growing season spans from late May to mid-September. Overall, ecosystem CO_2_ fluxes exhibited strong temporal variability, with higher fluxes observed during mid-growing season and under increased precipitation treatments.

**Figure 4 biology-14-01663-f004:**
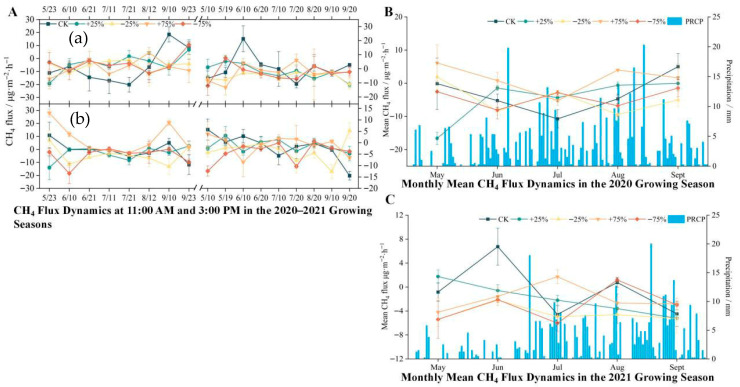
(**A**) Temporal and seasonal variation in ecosystem CH_4_ flux under different precipitation treatments during the 2020–2021 growing seasons. Diurnal CH_4_ flux dynamics at 11:00 a.m. (**a**) and 3:00 p.m. (**b**) from May to September in 2020 and 2021 under five precipitation treatments: control (CK), +25%, −25%, +75%, and −75%. Each point represents the mean flux of a specific sampling date, with error bars indicating the standard error of the mean (n = 3). (**B**) Monthly mean CH_4_ flux (line plots, left *Y*-axis) and daily precipitation (blue bars, right *Y*-axis) during the 2020 growing season. The flux values represent the average of all biweekly measurements within each month under each precipitation treatment. Error bars represent standard error (n = 6 per month). (**C**) Monthly mean CH_4_ flux and daily precipitation during the 2021 growing season, following the same structure as in panel B. The precipitation values in panels B and C are shared across all treatments as background climate conditions, while CH_4_ flux differs by treatment. All CH_4_ fluxes are expressed in mg·m^−2^·h^−1^. Precipitation is measured in mm. The growing season spans from late May to mid-September. Overall, ecosystem CH_4_ fluxes exhibited strong temporal variability, with higher fluxes observed during mid-growing season and under increased precipitation treatments.

**Figure 5 biology-14-01663-f005:**
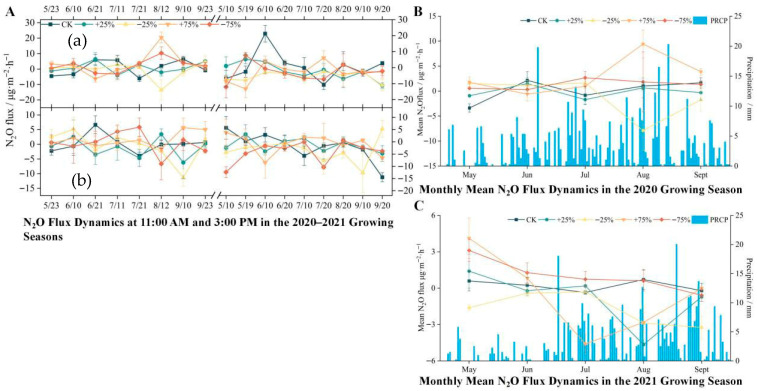
(**A**) Temporal and seasonal variation in ecosystem N_2_O flux under different precipitation treatments during the 2020–2021 growing seasons. Diurnal N_2_O flux dynamics at 11:00 a.m. (**a**) and 3:00 p.m. (**b**) from May to September in 2020 and 2021 under five precipitation treatments: control (CK), +25%, −25%, +75%, and −75%. Each point represents the mean flux of a specific sampling date, with error bars indicating the standard error of the mean (n = 3). (**B**) Monthly mean N_2_O flux (line plots, left *Y*-axis) and daily precipitation (blue bars, right *Y*-axis) during the 2020 growing season. The flux values represent the average of all biweekly measurements within each month under each precipitation treatment. Error bars represent standard error (n = 6 per month). (**C**) Monthly mean N_2_O flux and daily precipitation during the 2021 growing season, following the same structure as in panel B. The precipitation values in panels B and C are shared across all treatments as background climate conditions, while N_2_O flux differs by treatment. All N_2_O fluxes are expressed in mg·m^−2^·h^−1^. Precipitation is measured in mm. The growing season spans from late May to mid-September. Overall, ecosystem N_2_O fluxes exhibited strong temporal variability, with higher fluxes observed during mid-growing season and under increased precipitation treatments.

**Table 1 biology-14-01663-t001:** Vegetation Characteristics under Different Precipitation Treatments in the Growing Seasons of 2020 and 2021 in Wayan Mountain.

Year	Treatment	Dominant Species	Vegetation Cover (%)	Vegetation Height (cm)	Thickness of the Surface Vegetation Litter (cm)
2020	CK	*Kobresia tibetica*, *Carex* spp., *Saussurea pulchra*	84	5.4~12.6	0.21
	+25%	*Kobresia tibetica*, *Carex* spp., *Potentilla anserina*, *Saussurea pulchra*	92	6.3~17.4	0.16
	−25%	*Kobresia tibetica*, *Carex* spp., *Plantago asiatica*	82	4.9~15.3	0.28
	+75%	*Kobresia tibetica*, *Carex* spp., *Saussurea pulchra*, *Artemisia lactiflora*	90	7.1~17.8	0.37
	−75%	*Kobresia tibetica*, *Carex* spp., *Potentilla anserina*, *Lithospermum erythrorhizon*	80	0.9~13.7	0.23
2021	CK	*Kobresia tibetica*, *Carex* spp., *Saussurea pulchra*	85	5.6~11.8	0.23
	+25%	*Kobresia tibetica*, *Carex* spp., *Potentilla anserina*, *Saussurea pulchra*	93	7.4~17.5	0.31
	−25%	*Kobresia tibetica*, *Carex* spp., *Plantago asiatica*	81	5.2~13.6	0.25
	+75%	*Kobresia tibetica*, *Carex* spp., *Saussurea pulchra*, *Artemisia lactiflora*	87	8.2~16.8	0.28
	−75%	*Kobresia tibetica*, *Carex* spp., *Potentilla anserina*, *Lithospermum erythrorhizon*, *Rubus pileatus*	80	2.6~10.6	0.19

Nomenclature follows the Flora of China database.

**Table 2 biology-14-01663-t002:** Relationships between greenhouse gas fluxes and soil temperature and moisture.

	Treatment	CK			+25%			−25%			+75%			−75%		
	Gas	CO_2_	CH_4_	N_2_O	CO_2_	CH_4_	N_2_O	CO_2_	CH_4_	N_2_O	CO_2_	CH_4_	N_2_O	CO_2_	CH_4_	N_2_O
2020	ST	0.01	0.22	−0.02	0.06	0.41 **	−0.15	0.05	−0.06	−0.23	0.07	0.47 **	0.38 *	−0.22	0.05	−0.10
	SM	0.16	−0.58 **	−0.12	0.54 **	−0.27 *	−0.11	0.52 **	0.25	0.39 *	0.27 *	−0.35 *	−0.24	0.30 *	−0.11	−0.13
2021	ST	0.37 **	−0.17	−0.07	0.24	−0.11	−0.15	0.17	−0.01	−0.15	0.01	−0.11	−0.14	0.13	−0.15	−0.18
	SM	0.44 **	−0.21	−0.14	0.19	−0.33 *	−0.01	0.27 *	−0.04	0.38 **	0.26	0.06	−0.02	0.25	0.24	0.16

*: Indicates a *p*-value less than 0.05, meaning the result is significant at the 0.05 level. **: Indicates a *p*-value less than 0.01, meaning the result is significant at the 0.01 level.

**Table 3 biology-14-01663-t003:** Relationships between greenhouse gas fluxes and soil physicochemical factors under different precipitation treatments.

Treatment	Gas	2020				2021			
		TN (g/kg)	TC (g/kg)	pH	EC (uS/cm)	TN (g/kg)	TC (g/kg)	pH	EC (uS/cm)
CK	CO_2_	0.125	0.351 *	0.009	0.146	−0.226	0.321 *	−0.116	0.252
	CH_4_	−0.370 *	0.371 *	−0.175	−0.298	−0.215	−0.030	0.096	−0.127
	N_2_O	0.282	0.094	0.137	0.302	−0.398 *	−0.082	−0.803 **	−0.648 **
+25%	CO_2_	0.209	0.465 **	−0.171	0.033	−0.002	−0.042	0.433 *	−0.011
	CH_4_	−0.206	0.309	−0.206	0.151	−0.143	−0.120	−0.676 **	−0.326 *
	N_2_O	0.318 *	−0.209	−0.012	−0.289	−0.126	−0.158	−0.328 *	−0.090
−25%	CO_2_	0.262	0.422 *	−0.390 *	−0.080	0.531 **	0.575 **	0.211	0.098
	CH_4_	0.130	−0.003	−0.273	0.240	−0.221	−0.171	−0.146	−0.209
	N_2_O	0.260	−0.013	0.004	0.057	0.077	0.032	−0.356 *	−0.161
+75%	CO_2_	0.152	−0.213	−0.410 *	−0.062	0.240	0.075	0.326 *	0.336 *
	CH_4_	0.129	−0.067	0.030	0.034	−0.094	0.001	−0.002	−0.155
	N_2_O	−0.210	0.297	−0.431 *	0.682 **	−0.706 **	−0.509 **	−0.179	−0.108
−75%	CO_2_	0.059	0.026	0.051	−0.337 *	−0.014	0.278	0.084	−0.166
	CH_4_	−0.105	−0.305	−0.062	−0.022	0.062	0.073	0.398 *	0.194
	N_2_O	−0.119	−0.037	0.039	0.089	−0.209	−0.100	−0.165	−0.244

Notes: Values represent Pearson correlation coefficients (r). Positive and negative values indicate positive and negative correlations, respectively. *p* < 0.05 indicates statistical significance; *p* < 0.01 indicates high significance. *: Indicates a *p*-value less than 0.05, meaning the result is significant at the 0.05 level. **: Indicates a *p*-value less than 0.01, meaning the result is significant at the 0.01 level.

**Table 4 biology-14-01663-t004:** Vegetation conditions in the sample plots during the growing seasons of 2020 and 2021.

Year	Treatment	Plant Height /cm	Vegetation Cover /%	Aboveground Biomass /(g·m^−2^)	Underground Biomass /(g·m^−2^)	Simpson Index	Shannon Index	Pielou Evenness Index
2020	CK	5.4~12.6	84	80.11	2341.52	0.022	0.670	0.322
	+25%	6.3~17.4	92	145.90	2591.75	0.025	0.931	0.448
	−25%	4.9~15.3	82	103.488	2046.81	0.307	0.839	0.403
	+75%	7.1~17.8	90	140.69	3002.22	0.012	0.766	0.368
	−75%	0.9~13.7	80	102.98	1860.27	0.011	0.731	0.351
2021	CK	5.6~11.8	85	100.74	1981.09	0.090	0.809	0.389
	+25%	7.4~17.5	93	153.19	2040.23	0.035	0.837	0.402
	−25%	5.2~13.6	81	109.14	1823.88	0.317	0.834	0.401
	+75%	8.2~16.8	87	150.63	2195.43	0.063	0.883	0.425
	−75%	2.6~10.6	80	100.96	1462.44	0.086	0.971	0.425

## Data Availability

The data presented in this study are available on request from the corresponding author due to privacy or ethical restrictions.

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
