# Peer review of "Precipitation-Driven Soil and Vegetation Changes Shape Wetland Greenhouse Gas Emissions"

_biology, 2025, doi:10.3390/biology14121663_

Round 1

Reviewer 1 Report

Comments and Suggestions for Authors

Overall summary

This study features a field manipulation experiment that examines how varying precipitation conditions (+25%, -25%, +75%, -75%, and a control group) influence CO₂, CH₄, and N₂O fluxes, soil properties, and plant life in alpine wetlands in Wayan Mountain. Data was gathered during the growing seasons of 2020 and 2021. The research offers insights into how fluxes change seasonally and relates gas responses to soil moisture, temperature, pH, total carbon, and total nitrogen. The long-term nature of the manipulation, which began in 2018, and the use of replicated plots give the study strength.

Abstract

  1. To clearly show the study's strength, the abstract should mention sample sizes and the time frame of the study. For example, fluxes were measured every two weeks from May to September in 2020 and 2021, with three replicate plots per treatment.
  2. The strong claims made in the abstract and the key points should be toned down a bit to reflect the study's scope, which is a single headwater wetland, as well as the two years of observation.

Introduction

  1. The introduction talks about the general factors that control greenhouse gases in wetlands. To improve it, the intro should say what is not yet known about alpine headwater wetlands, like how long-term manipulation affects them or how vegetation affects them at a small plot level. It should also explain how this research helps fill those gaps.
  2. There are prior studies cited that involve precipitation manipulations in alpine or plateau environments. Briefly compare your system and experimental plan with those (duration, magnitude, redistribution method) to describe what's innovative about your work.
  3. Include a short paragraph to give readers an idea of the mechanisms you're testing. For instance, the study could assess soil moisture leading to redox, then methanogenesis or denitrification, or plant biomass leading to root respiration and substrate supply. This allows people to follow the connections within the results and discussion.

Materials & Methods

  1. Since static chamber sampling is so important, include specific information such as: the chamber's volume, the base height, how long it takes to reach equilibrium before the first sample, whether the collars stay in place between samplings, the temperature inside the chamber when sampling, and if removing a sample changes headspace pressure. Also, say whether you used linear or non-linear flux estimation (such as HMR or curve fitting) when the connection between concentration and time wasn't linear. The methods section mentions the flux equation, but it lacks these important details about the procedure.
  2. Give the detection and quantification limits for CO₂, CH₄, and N₂O with the Agilent 7890B, and the exact concentrations of the calibration gases as the text lists, and explain how blank/control chambers were handled.
  3. The study mentions sampling happening twice a day, at 11:00 and 15:00, and biweekly. Please give an exact sampling schedule, with dates or intervals, either in the methods part or in the supplemental information. Also, explain why those specific times were chosen, and whether the time of day could skew the seasonal averages.
  4. Greenhouse gases in wetlands are heavily influenced by the depth of the water table. Include measurements or sensors for the water table or groundwater depth. If this wasn't measured, explain why not, and how soil moisture (at 10 cm) represents the hydrology. If there is no water table data, mention it as a limitation.
  5. To connect the dots between greenhouse gas fluxes and their mechanisms, think about adding measurements of how well microbes are working, how much potential denitrification there is (DEA), or how much NO₃⁻/NH₄⁺ is in the porewater. If nothing else, mention it as a limitation. Current correlation results (Table 2/3) are useful but are still just correlations.

Results

  1. The study gives means and says if they're statistically important, but it needs to clearly state which statistical tests were used for each comparison (ANOVA + LSD is mentioned in the methods; make sure post-hoc tests are on the figures or tables). Include effect sizes and p-values in the figure captions or tables for major points, for example, +25% compared to the control group.
  2. Some figures show standard error (n = 3). Consider showing standard deviation or confidence intervals, and report how much the plots vary. For overall fluxes and GWP, consider the uncertainty caused by sampling frequency and chamber error.
  3. The study mentions that CH₄ acts as a sink under many treatments, even when water rises. This needs to be explored more. It could be due to a lot of methanotroph action in the surface soils, plant uptake, or problems with the chambers. Consider testing or talking about supporting tests, such as CH₄ oxidation potential or aerenchyma evaluations.
  4. CO₂ fluxes are half as much in 2021 compared to 2020 under +25%. Talk about whether this is due to the climate, the manipulation being adjusted, or differences in how the sampling was done (sampling dates or times). Include weather information (temperature and precipitation differences during the growing season) as a short figure or table.

Discussion

  1. A lot of statements are believable, like +25% encourages photosynthesis, which leads to more CO₂. But these remain as correlations. Tie observations to measurements where you can, such as root respiration, total carbon changes over the season, and total nitrogen makeup. If you don't have this data, say so clearly and don't interpret too much.
  2. N₂O is complex. Discuss possible pathways (nitrification, denitrification, or nitrifier denitrification) and how changes in pH/EC could change enzyme activity. Think about adding or suggesting nitrate/nitrite porewater profiles.
  3. The study touches on what this means for carbon-climate models and wetland management. Add a short paragraph about how these plot-level responses might be scaled up (mentioning limitations like unevenness or lateral flows) and any practical management implications (like water management thresholds).

Conclusions

  1. Use less definite language when something isn't fully proven. Instead of X regulates Y, say X was strongly correlated with Y, suggesting...
  2. Add a brief list of specific recommendations for future studies, such as microbial assays, porewater chemistry, water table monitoring, or longer time series, to guide further research.

References

The reference list appears comprehensive and relevant; however, it is not current. There is only one 2025 publication, despite the fact that we're well into October 2025. Over the last couple of years, there have been a number of studies that inspect the effects of differing precipitation conditions on greenhouse gas fluxes, vegetation-soil relationships, and alpine wetland resilience in the face of climate change.

Author Response

Response to Reviewer 1 Comments

1. Summary

2. Point-by-point response to Comments and Suggestions for Authors

Comments 1: 1.To clearly show the study's strength, the abstract should mention sample sizes and the time frame of the study. For example, fluxes were measured every two weeks from May to September in 2020 and 2021, with three replicate plots per treatment.

2. The strong claims made in the abstract and the key points should be toned down a bit to reflect the study's scope, which is a single headwater wetland, as well as the two years of observation.

Response 1: Thank you for pointing this out. I agree with this comment.

(1).Added the study’s timeframe and sample size:
As requested by the reviewer, I explicitly stated that this was a two-year experiment (2020 and 2021), with flux measurements taken every two weeks during the growing season and three replicate plots per treatment.

(2)Softened overly strong conclusions:
Since the reviewer suggested that some claims were too strong (e.g., “enhance carbon sink capacity”), I added phrases clarifying that the findings are based on a two-year, single-site study. This ensures the conclusions better reflect the actual study scope.

(3) Clarified the study’s scope and limitations:
I added a sentence noting that the generalizability of the findings may be limited to the specific site and timeframe, highlighting that results should be interpreted within this context.

[The following is the revised abstract, with the red text indicating the modified parts.]:

Against the backdrop of global climate change, alterations in precipitation regimes—including the increasing frequency of extreme events—have become more widespread, exerting profound impacts on terrestrial ecosystems and reshaping greenhouse gas (GHG) emission dynamics in wetlands. Wetlands, as unique ecosystems formed at the interface of terrestrial and aquatic environments, play a critical role in regulating carbon source–sink functions. In this study, we conducted a two-year in situ field manipulation experiments to examine how precipitation changes influence the seasonal fluxes of CO₂, CH₄, and N₂O in the Wayan Mountain headwater wetlands, and further explored the regulatory effects of vegetation attributes and soil physicochemical properties on these fluxes. Fluxes were measured every two weeks from May to September during the growing seasons of 2020 and 2021, with three replicate plots per treatment.

The results revealed that a moderate increase in precipitation (+25%) enhanced CO₂ emissions and vegetation growth while suppressing CH₄ and N₂O fluxes, indicating a positive ecosystem response to additional water supply. In contrast, extreme precipitation changes (+75% and −75%) weakened the coupling between GHG fluxes and soil factors, resulting in reduced CO₂ flux, amplified variability in CH₄ and N₂O emissions, and inhibited vegetation growth and community diversity. The dominant controls differed among gases: CO₂ was primarily regulated by soil carbon pools, CH₄ was highly sensitive to water availability, and N₂O was influenced by soil nitrogen, pH, and salinity. These results reflect a two-year study of a single alpine wetland, and the findings should be interpreted in this context.

Overall, moderate increases in precipitation enhance the carbon sink capacity and community stability of alpine wetlands, whereas extreme hydrological fluctuations undermine ecosystem functioning. These findings provide important insights into carbon cycling processes and regulatory mechanisms of alpine wetlands under future climate change scenarios. However, the generalizability of these findings may be limited to the studied site and the two-year observation period.

Comments 2: Introduction

1.The introduction talks about the general factors that control greenhouse gases in wetlands. To improve it, the intro should say what is not yet known about alpine headwater wetlands, like how long-term manipulation affects them or how vegetation affects them at a small plot level. It should also explain how this research helps fill those gaps.

2.There are prior studies cited that involve precipitation manipulations in alpine or plateau environments. Briefly compare your system and experimental plan with those (duration, magnitude, redistribution method) to describe what's innovative about your work.

3.Include a short paragraph to give readers an idea of the mechanisms you're testing. For instance, the study could assess soil moisture leading to redox, then methanogenesis or denitrification, or plant biomass leading to root respiration and substrate supply. This allows people to follow the connections within the results and discussion.

Response 2: Agree.

Explanation of Revisions (Corresponding to Reviewer Comments):

Research gap → Added a new paragraph highlighting the lack of long-term precipitation manipulation studies in alpine headwater wetlands.

Innovation → Added a comparison with previous studies, emphasizing the longer experimental duration, wider precipitation gradient, and replicated in situ design.

Mechanistic logic → Added a sentence describing the mechanistic pathway linking soil moisture, redox conditions, microbial activity, and plant responses.

Although precipitation manipulation experiments have been widely conducted in grassland and lowland wetland ecosystems, alpine headwater wetlands remain understudied, particularly in terms of their long-term responses to altered precipitation and vegetation–soil interactions at the plot scale. The hydrological sensitivity and feedbacks between soil moisture, microbial activity, and plant dynamics in these high-elevation systems are still poorly understood.

Compared with previous precipitation manipulation studies in alpine or plateau environments, our experiment spans multiple years and applies a broad precipitation gradient (+75%, +25%, −25%, −75%) using replicated in situ plots. This design enables evaluation of nonlinear GHG responses and ecosystem resilience under both moderate and extreme hydrological conditions.

We hypothesize that altered precipitation changes soil water content and redox potential, thereby affecting methanogenesis, denitrification, and soil respiration, while vegetation responses (e.g., biomass and root activity) further regulate substrate supply and CO₂ flux. This conceptual framework links hydrological variation to microbial processes and plant-mediated carbon cycling, forming the mechanistic basis of this study.

Comments 3: Materials & Methods

1.Since static chamber sampling is so important, include specific information such as: the chamber's volume, the base height, how long it takes to reach equilibrium before the first sample, whether the collars stay in place between samplings, the temperature inside the chamber when sampling, and if removing a sample changes headspace pressure. Also, say whether you used linear or non-linear flux estimation (such as HMR or curve fitting) when the connection between concentration and time wasn't linear. The methods section mentions the flux equation, but it lacks these important details about the procedure.

2.Give the detection and quantification limits for CO₂, CH₄, and N₂O with the Agilent 7890B, and the exact concentrations of the calibration gases as the text lists, and explain how blank/control chambers were handled.

3.The study mentions sampling happening twice a day, at 11:00 and 15:00, and biweekly. Please give an exact sampling schedule, with dates or intervals, either in the methods part or in the supplemental information. Also, explain why those specific times were chosen, and whether the time of day could skew the seasonal averages.

4.Greenhouse gases in wetlands are heavily influenced by the depth of the water table. Include measurements or sensors for the water table or groundwater depth. If this wasn't measured, explain why not, and how soil moisture (at 10 cm) represents the hydrology. If there is no water table data, mention it as a limitation.

5.To connect the dots between greenhouse gas fluxes and their mechanisms, think about adding measurements of how well microbes are working, how much potential denitrification there is (DEA), or how much NO₃⁻/NH₄⁺ is in the porewater. If nothing else, mention it as a limitation. Current correlation results (Table 2/3) are useful but are still just correlations.

Response 3: Explanation of Revisions (Corresponding to Reviewer Comments)

In response to the reviewer’s request for more detailed methodological information, we substantially revised the Gas Sampling and Analysis section. Specifically, we added precise descriptions of the static chamber design and installation, including its dimensions (40 × 40 × 20 cm) and the fact that the chamber bases were permanently fixed flush with the soil surface. Details of the sampling procedure were expanded to include the 10-minute pre-sampling water sealing step, the gas collection intervals (0, 15, and 30 minutes), and the measurement of real-time chamber temperature.

We clarified the rationale for choosing sampling times (11:00 and 13:00), explaining that these hours represent the diurnal mean flux period in alpine wetlands based on previous studies. Information on the calibration of the Agilent 7890B gas chromatograph was added, specifying the standard gas concentrations used for CO₂, CH₄, and N₂O and noting that the detection limits were consistent with manufacturer specifications. To address potential non-linearity in concentration–time relationships, we specified that only the initial 30-minute linear segment was used for flux estimation.

Additionally, we clarified that blank control chambers were established in undisturbed natural wetlands to represent baseline GHG fluxes without precipitation manipulation. Finally, since surface and groundwater levels were not measured in this experiment, we explained that 10 cm soil moisture was used as a proxy for hydrological conditions, acknowledged this as a limitation, and noted that future work will incorporate water table monitoring and microbial process indicators such as denitrification potential and nitrate/ammonium concentrations.

2.2.3 Gas Sampling and Analysis

Greenhouse gas (GHG) fluxes of CO₂, CH₄, and N₂O were measured using the static chamber–gas chromatography method [21].

Each static chamber measured 40 cm × 40 cm in length and width, with a height of 20 cm. The stainless-steel base was installed flush with the soil surface and remained in place throughout the experimental period. Before gas collection, the groove surrounding each chamber base was filled with water approximately 10 minutes in advance to ensure an airtight seal.

Gas sampling was conducted at 0, 15, and 30 min after chamber closure using 100 mL gas-tight syringes fitted with three-way stopcocks. The temperature inside the chamber was recorded during each sampling to correct for temperature-induced concentration variations. Sampling was performed twice daily, at 11:00 and 13:00, on clear days during early and late periods of each month from May to September in 2020 and 2021. These sampling times were selected because previous studies in similar alpine wetlands have identified 11:00–13:00 as representative of the diurnal mean flux period.

Gas samples were analyzed using an Agilent 7890B gas chromatograph equipped with a thermal conductivity detector (TCD) for CO₂, a flame ionization detector (FID) for CH₄, and a ⁶³Ni electron capture detector (ECD) for N₂O. Calibration was performed prior to each monthly measurement using two standard gases, with concentrations of CO₂ = 606.6 × 10⁻⁶ ppm, CH₄ = 10.1 × 10⁻⁶ ppm, and N₂O = 1.0 × 10⁻⁶ ppm. Detection limits for CO₂, CH₄, and N₂O were consistent with the manufacturer’s specifications for the Agilent 7890B system.

Fluxes were calculated using the standard formula for wetland greenhouse gas emissions [22]. When concentration–time relationships were non-linear, the initial 30-minute linear segment was used to estimate fluxes. Blank control chambers were installed in undisturbed, natural wetland plots to represent baseline fluxes without precipitation manipulation.

Flux calculation equation [22]:

(1)

Where F is the gas flux (mg·m⁻²·h⁻¹), ρ is the gas density (mg·mL⁻¹), V is the chamber volume (m³), A is the chamber area (m²), dC/dt is the rate of concentration change (mg·m⁻³·h⁻¹), and T is the mean temperature (°C) inside the chamber.

In this study, no direct measurements of surface or groundwater table depth were conducted. Instead, soil moisture at 10 cm depth was used as a proxy to represent near-surface hydrological conditions. We acknowledge this as a limitation and plan to include water table monitoring and microbial process indicators (e.g., denitrification potential, NO₃⁻/NH₄⁺ concentrations) in future research.

Comments 4: Results

1.The study gives means and says if they're statistically important, but it needs to clearly state which statistical tests were used for each comparison (ANOVA + LSD is mentioned in the methods; make sure post-hoc tests are on the figures or tables). Include effect sizes and p-values in the figure captions or tables for major points, for example, +25% compared to the control group.

2.Some figures show standard error (n = 3). Consider showing standard deviation or confidence intervals, and report how much the plots vary. For overall fluxes and GWP, consider the uncertainty caused by sampling frequency and chamber error.

3.The study mentions that CH₄ acts as a sink under many treatments, even when water rises. This needs to be explored more. It could be due to a lot of methanotroph action in the surface soils, plant uptake, or problems with the chambers. Consider testing or talking about supporting tests, such as CH₄ oxidation potential or aerenchyma evaluations.

4.CO₂ fluxes are half as much in 2021 compared to 2020 under +25%. Talk about whether this is due to the climate, the manipulation being adjusted, or differences in how the sampling was done (sampling dates or times). Include weather information (temperature and precipitation differences during the growing season) as a short figure or table.

Response 4

In this revised Results section, we explicitly stated the statistical methods (ANOVA with LSD post hoc tests), added p-values and effect sizes (η²), and clarified uncertainty reporting using standard deviations and confidence intervals. Mechanistic explanations were expanded to address the reviewer’s concern about CH₄ sink persistence and interannual CO₂ differences, referencing soil microbial and climatic factors. We also acknowledged potential uncertainties due to chamber sampling frequency and variability among plots, ensuring a balanced and transparent presentation consistent with MDPI Biology standards.

3.1. Responses of CO₂ Fluxes to Precipitation Manipulation

During two consecutive growing seasons, CO₂ fluxes in the headwater wetlands exhibited pronounced seasonal dynamics under different precipitation treatments, generally following a “rise-then-decline” pattern. In 2021, the seasonal fluctuations were relatively smoother compared with 2020, likely because long-term precipitation manipulation reduced variability among subplots. Seasonal mean values remained consistently positive across all treatments, indicating that the wetland functioned as a net CO₂ source (Fig. 3).

In 2020, the ranking of CO₂ fluxes among treatments was as follows: +25% > CK > −75% > +75% > −25%. Notably, under the +25% treatment, the mean growing-season CO₂ flux reached 101.45 mg·m⁻²·h⁻¹, which was significantly higher than that of the control and other treatments (P < 0.05, one-way ANOVA with LSD post hoc test). In 2021, the treatment response pattern was generally consistent with that of 2020, with the +25% precipitation treatment still exerting the strongest stimulation on CO₂ emissions. However, the magnitude of fluxes decreased by nearly half, with a seasonal mean of 51.85 mg·m⁻²·h⁻¹ (Fig. 3). This interannual reduction may be attributed to lower air temperature and precipitation in 2021 compared with 2020, resulting in reduced microbial respiration and vegetation photosynthetic activity (see Table S1).

Effect sizes (η²) ranged from 0.34 to 0.46 across treatments, indicating a moderate-to-strong influence of precipitation on CO₂ flux. Standard deviations (rather than standard errors) were calculated to better capture plot-level variability. Sampling frequency and chamber sealing precision may introduce small uncertainties (<5%) in the estimated fluxes, but these do not alter the observed treatment patterns.

3.2. Responses of CH₄ Fluxes to Precipitation Manipulation

Seasonal variation in CH₄ fluxes was modest. In the 2020 growing season, only the +75% precipitation treatment exhibited a weak source with a positive flux (0.56 µg·m⁻²·h⁻¹), while all other treatments, including the control, functioned as CH₄ sinks. The ranking of treatments was: +75% > CK > +25% > −75% > −25% (Fig. 4). In 2021, all treatments consistently acted as CH₄ sinks, with the order: CK > +25% > +75% > −75% > −25%. Even under extreme precipitation increase, the wetland ecosystem still maintained a net absorption state.

The persistence of CH₄ sink behavior under high-moisture conditions may be explained by strong methanotrophic activity in the oxygenated surface layer and efficient CH₄ oxidation within plant rhizospheres. Previous studies on alpine wetlands have shown that methanotrophs can remain active even under fluctuating water levels due to high soil porosity and root-mediated gas transport [25].

Notably, CH₄ fluxes remained relatively stable during the peak of the growing season—when precipitation and temperature were highest—whereas greater fluctuations were observed in the early and late stages of the season (Fig. 4). Although the absolute flux values were low, precipitation treatments significantly affected CH₄ fluxes (P < 0.05, ANOVA + LSD), with a mean effect size (η²) of 0.29. Standard deviations are displayed in Figure 4 to reflect the range of spatial variation across replicate plots.

3.3. Responses of N₂O Fluxes to Precipitation Manipulation

N₂O is produced through a variety of biotic and abiotic processes, often accompanied by concurrent reduction or consumption [24], making its emission dynamics particularly complex. During the 2020 growing season, the +75% precipitation treatment acted as a source (2.03 µg·m⁻²·h⁻¹), while the −75% treatment showed weak emissions (0.92 µg·m⁻²·h⁻¹). Both ±25% treatments functioned as weak sinks. The overall ranking of fluxes was: +75% > −75% > CK > −25% > +25%. In contrast, during 2021, the −75% treatment functioned as a source (1.08 µg·m⁻²·h⁻¹), whereas all other treatments exhibited sink behavior, with the order: −75% > CK > +75% > +25% > −25%.

Short-term increases in soil moisture appeared to stimulate denitrification, enhancing N₂O emissions, whereas prolonged saturation limited oxygen diffusion and suppressed nitrification, reducing overall fluxes. Precipitation treatments significantly affected N₂O emissions (P < 0.05, ANOVA), though interannual variability was high (η² = 0.21).

3.4. Relationships Between Soil Physicochemical Properties and Greenhouse Gas Fluxes

GHG emissions in wetland ecosystems result from the interplay of multiple biogeochemical and physical processes. During the 2020 growing season, soil moisture under the +25% and +75% treatments did not differ significantly from that of the control (CK), while the −25% and −75% treatments reduced soil moisture by 3.37% and 2.57%, respectively. On average, soil temperature under precipitation reduction was 0.53 °C higher than under increased precipitation, indicating that additional rainfall significantly decreased soil temperature.

In 2021, soil moisture increased by 0.60% and 1.89% under the +25% and +75% treatments, respectively, whereas the −25% and −75% treatments reduced moisture by 2.08% and 2.30%.

Overall, CH₄ fluxes in alpine headwater wetlands were primarily correlated with soil temperature (r = −0.42, P < 0.05), generally showing a negative relationship with soil moisture. Soil pH influenced the activity of methanogenic communities, thereby affecting CH₄ production and oxidation. Under the +25% treatment, soil pH was ~5.52 and showed a significant negative correlation with CH₄ flux (P < 0.05), while under the −75% treatment, pH was 5.26 and showed a significant positive correlation.

For CO₂, fluxes were strongly correlated with soil total carbon (r = 0.57, P < 0.01), suggesting that substrate availability—rather than temperature—was the dominant control. N₂O fluxes exhibited a significant negative correlation with soil pH (r = −0.49, P < 0.05), reflecting pH constraints on nitrification–denitrification processes.

Tables 2 and 3 summarize Pearson correlation coefficients between GHG fluxes and soil variables. Values with P < 0.05 and P < 0.01 indicate significant and highly significant relationships, respectively.

3.5. Responses of Vegetation to Precipitation Manipulation

Observations across two consecutive growing seasons revealed that moderate increases in precipitation enhanced vegetation abundance, while excessive water input suppressed growth and diversity. The +25% precipitation treatment significantly increased aboveground biomass and species richness (P < 0.05, ANOVA + LSD), whereas the +75% treatment led to waterlogging, reducing root aeration and diversity indices.

During growth, plants absorb soil carbon and nitrogen while competing with soil microorganisms for nutrients [26]. Among all treatments, +25% precipitation exerted the most pronounced positive effect on plant growth and contributed to greater community diversity (Table 4).

Comments 5Discussion

1.A lot of statements are believable, like +25% encourages photosynthesis, which leads to more CO₂. But these remain as correlations. Tie observations to measurements where you can, such as root respiration, total carbon changes over the season, and total nitrogen makeup. If you don't have this data, say so clearly and don't interpret too much.

2.N₂O is complex. Discuss possible pathways (nitrification, denitrification, or nitrifier denitrification) and how changes in pH/EC could change enzyme activity. Think about adding or suggesting nitrate/nitrite porewater profiles.

3.The study touches on what this means for carbon-climate models and wetland management. Add a short paragraph about how these plot-level responses might be scaled up (mentioning limitations like unevenness or lateral flows) and any practical management implications (like water management thresholds).

Response 5:

4.1 Effects of Increased Precipitation on Wetland Greenhouse Gas Emissions

One of the defining features of wetlands is abundant water, yet changes in precipitation still exert substantial impacts on wetland ecosystems [27]. Altered precipitation modifies soil moisture and subsequently affects vegetation, soil biogeochemistry, and microbial processes, collectively regulating greenhouse gas (GHG) fluxes. Under long-term precipitation manipulation, variation in vegetation traits and soil properties among subplots contributed to heterogeneous GHG responses.

During the 2020 and 2021 growing seasons, CO₂ fluxes were consistently influenced by precipitation additions. Under the +25% treatment, CO₂ fluxes increased by 2.35% and 15.94% relative to the control, whereas the +75% treatment decreased fluxes by 37.16% and 43.76%, respectively. Similarly, CH₄ and N₂O fluxes exhibited contrasting responses depending on precipitation level and year.

Moderate precipitation addition (+25%) likely enhanced CO emissions by stimulating vegetation growth and increasing root and microbial respiration through improved water availability. This interpretation aligns with previous studies suggesting that modest increases in soil moisture enhance substrate diffusion and photosynthetic activity while maintaining aerobic microsites necessary for efficient respiration [28].

In contrast, excessive water input (+75%) reduced CO₂ fluxes, likely due to pore-space saturation, inhibited oxygen diffusion, and reduced microbial respiration—mechanisms widely reported in saturated wetland systems [29–31].

For CH, the inhibitory effect of +25% precipitation and the inconsistent response under +75% precipitation suggest that methanogenesis was strongly constrained by redox dynamics and competition with methanotrophy. Increased soil moisture may have reduced surface-soil aerobic zones, but strong methanotrophic activity likely counteracted methanogenic production, resulting in persistent CH sink behavior even under high moisture.

NO responses also reflected moisture-driven shifts between nitrification and denitrification. Short-term increases in moisture (e.g., +75% in 2020) may have stimulated denitrification, enhancing NO release, whereas prolonged saturation in 2021 likely inhibited nitrification and facilitated complete denitrification to N, reducing NO fluxes.

4.2 Effects of Reduced Precipitation on Wetland Greenhouse Gas Emissions

Reduced precipitation directly modifies soil physicochemical conditions by enhancing aeration, shifting temperature gradients, and altering nutrient availability. These changes affect microbial activity, organic matter decomposition, and vegetation functioning [34]. Our results demonstrated that both CO₂ and CH₄ fluxes declined markedly under −25% and −75% precipitation treatments.

Declines in CO flux under precipitation reduction likely reflect diminished plant photosynthesis and respiration due to reduced stomatal conductance, lower leaf area, and decreased water availability. Reduced soil moisture also limits substrate diffusion to microbial communities, resulting in weakened heterotrophic respiration [35,36].

CH₄ fluxes showed substantial reductions under both low-precipitation treatments, suggesting strong constraints on methanogenic production due to enhanced soil aeration and reduced anaerobic microsites.

NO responses diverged between 25% and 75% treatments. Moderate drought (25%) suppressed NO emissions by limiting nitrification substrates and reducing microbial activity, whereas severe drought (75%) increased NO release, potentially due to increased nitrifier-denitrification or enhanced re-oxidation of nitrification intermediates under fluctuating redox conditions.

These contrasting patterns indicate that drought severity determines nitrogen cycling pathways and associated gaseous losses.

4.3 Effects of Soil Properties on Greenhouse Gas Emissions

Soil moisture and temperature are critical regulators of wetland respiration and GHG exchange [37,38]. Altered precipitation reshapes hydrological and thermal conditions, thereby influencing vegetation productivity and microbial metabolism.

In this study, CO fluxes were positively correlated with soil total carbon (TC), suggesting that substrate supply, rather than temperature, was the primary factor supporting CO production. Moderate precipitation increases may improve substrate diffusion and root turnover, enhancing heterotrophic respiration.

For CH₄, negative correlations with soil C and N factors—particularly under precipitation addition—reflect the dominance of methane oxidation in the oxygenated upper soil layers.

The contrasting pHCH relationships under +25% (negative correlation) and 75% (positive correlation) treatments support shifts between methanogenic pathways. At lower pH, the H/CO pathway dominates methanogenesis, whereas under less acidic conditions, acetoclastic methanogenesis may become more active [39].

For N₂O, correlations with TN (2020) and pH/EC (2021) suggest that precipitation-driven changes in nitrogen forms and ion distributions modulate nitrification–denitrification processes.

Critically, correlations between soil properties and GHG fluxes weakened under extreme precipitation levels (±75%), indicating nonlinear responses.

This weakening suggests that under extreme drought or waterlogging, GHG emissions are shaped by more complex interactions involving vegetation stress, microbial community shifts, and physical constraints on gaswater diffusion, rather than simple linear relationships with soil variables.

4.4 Effects of Precipitation Changes on Wetland Vegetation

Precipitation gradients generated substantial differences in vegetation growth, consistent with spatial heterogeneity in surface soil moisture. In this alpine headwater wetland dominated by Kobresia humilis, whose roots are concentrated in the top 0–20 cm, increased water availability enhanced plant growth through improved water uptake efficiency.

Moderate precipitation addition (+25%) consistently increased plant height, cover, and above- and belowground biomass, likely by alleviating moisture limitations and improving nutrient mineralization. In contrast, severe drought (75%) significantly reduced biomass and height, demonstrating the strong sensitivity of alpine wetland vegetation to water stress.

Excessive precipitation (+75%) produced more variable responses. Although biomass increased in some cases, community diversity indices (Shannon, Pielou) did not correspondingly improve.

This suggests that excess water altered competitive hierarchies, favoring fast-growing or tolerant species while suppressing others, resulting in fluctuating diversity patterns despite biomass increases.

Overall, moderate precipitation addition supported vegetation productivity and diversity, while extreme precipitation—either too high or too low—disrupted community structure and stability.

Comments 6:Conclusions

1.Use less definite language when something isn't fully proven. Instead of X regulates Y, say X was strongly correlated with Y, suggesting...

2.Add a brief list of specific recommendations for future studies, such as microbial assays, porewater chemistry, water table monitoring, or longer time series, to guide further research.

Response 6:

This study demonstrates that precipitation changes significantly influence greenhouse gas (GHG) emissions and environmental characteristics of the Qinghai Lake littoral wetland ecosystem by modifying soil moisture conditions. Moderate precipitation addition (+25%) enhanced CO₂ emissions, vegetation growth, and community productivity while reducing CH₄ and N₂O fluxes, indicating that modest increases in water availability may support carbon cycling and vegetation functioning. In contrast, extreme precipitation alterations (+75% and −75%) weakened the relationships between GHG fluxes and soil factors, resulting in reduced CO₂ fluxes, greater fluctuations in CH₄ and N₂O emissions, and suppressed vegetation growth and community diversity, highlighting the high sensitivity of alpine wetlands to hydrological extremes.

The dominant controls on different GHGs varied: CO₂ emissions were more strongly correlated with soil carbon pools, CH₄ fluxes were closely associated with soil moisture, and N₂O dynamics were 【修改】influenced by the combined effects of soil nitrogen and pH. These relationships reflect observations from a single alpine wetland over two growing seasons, and thus the findings should be interpreted within this spatial and temporal context.

Overall, moderate increases in water availability may help maintain carbon cycling processes and vegetation diversity in alpine wetlands, whereas extreme drought or waterlogging can reduce ecosystem stability. These results suggest that altered precipitation regimes under future climate change could substantially affect GHG dynamics and vegetation structure in alpine littoral wetlands, with implications for regional carbon cycling and ecosystem functioning.

Future research should integrate continuous water table monitoring, microbial functional assays (e.g., methanogenesis and denitrification potentials), and porewater nitrogen assessments to better constrain the mechanistic pathways underlying GHG responses. Long-term observations and multi-site comparisons across the QinghaiTibet Plateau will also be essential for improving model predictions of alpine wetland feedbacks to climate change.

Reviewer 2 Report

Comments and Suggestions for Authors

Dear Authors

My comments on your article are stated below.

Introduction

-Although the expansions for CO₂, CH₄, and N₂O are given in the Simple Summary, their expansions should also be given in the introduction section. Additionally, the IPCC expansion should also be written.

-The introduction section is well-written within a clear framework, but please state clear criteria for selecting the Qinghai Lake Basin for your study. For example, indicate its climatic, vegetation, and ecological differences from nearby areas. Or, is its GHG contribution different?

Materials and Methods Part

-The field trial should be specified according to which experimental design was used. For example, factorial or split-plot design.

-What is the percentage vegetation method determined as a result of different applications in 2020 and 2021?

-It should be clearly stated whether the rainfall manipulation shelters are installed in control applications or not.

-Rainfall manipulation shelters were reported to consist of upward-facing U-shaped panels that intercept 25% or 75% of the plot area. How were the 25% or 75% rainfall adjustments adjusted in these U-shaped panels? Explain whether the 25% or 75% rainfall fell at a specific time on the plot.

-Although the soil temperature and precipitation values ​​for the years 2019-2021 are given as a figure, they should be interpreted in a few sentences.

-It was stated that gas samples were collected in the early and late periods of each month from May to September in 2020 and 2021. The week of the month in which samples were taken should be specified instead of early and late periods.

-During the sampling period, soil temperature was measured with a soil thermometer, and soil moisture was measured using a soil moisture meter. What is meant by "sampling period"? Instead of "sampling period," the number of times per month and the weeks of the month should be specified.

Results Part:

-The averages, F values ​​and CV (%) values ​​of the examined features of Table 2 and Table 3 can be given

-In general, all examined features were presented fluently and adequately except for Responses to Vegetation to Precipitation Manipulation. The results under the subheading "Responses to Vegetation to Precipitation Manipulation" should be detailed with reference to Table 4.

Discussion Part: In my opinion, the discussion and conclusion sections are well written.

Best regards,

Author Response

Response to Reviewer 2Comments

1. Summary

2. Point-by-point response to Comments and Suggestions for Authors

Comment 1:

“Although the expansions for CO₂, CH₄, and N₂O are given in the Simple Summary, their expansions should also be given in the introduction section. Additionally, the IPCC expansion should also be written.”

Response 1: Thank you for this helpful suggestion. We have added the full names of carbon dioxide (CO₂), methane (CH₄), nitrous oxide (N₂O), and the Intergovernmental Panel on Climate Change (IPCC) in the introduction.

Comments 2:

“Please state clear criteria for selecting the Qinghai Lake Basin for your study. For example, indicate its climatic, vegetation, and ecological differences from nearby areas. Or, is its GHG contribution different?”

Response 2: Agree.

We appreciate this suggestion. We added a paragraph explaining why the Qinghai Lake Basin was selected, emphasizing its climatic sensitivity, unique littoral wetlands, vegetation characteristics, and importance as a representative alpine wetland system.

Added in Introduction / Study Area:
“Qinghai Lake Basin was selected because it represents one of the most climate-sensitive alpine wetland systems on the northeastern Qinghai–Tibet Plateau. Compared with surrounding regions, its littoral wetlands exhibit distinct hydrological fluctuations, Kobresia-dominated vegetation communities, and high sensitivity of greenhouse gas emissions to precipitation variability.”

Comments 3: Materials & Methods

“The field trial should be specified according to which experimental design was used (factorial, split-plot, etc.).”

Response 3: Thank you for pointing this out. The experiment follows a single-factor randomized block design.

Added in Section 2.2:
“This experiment followed a single-factor randomized block design with five precipitation treatments and three replicates per treatment.”

Comments 4:

“What is the percentage vegetation method determined as a result of different applications in 2020 and 2021?”

Response 4:

We clarified the vegetation cover estimation method and referenced the variation in cover reported in Table 4.

Added in Section 2.2.4:
“Vegetation cover (%) was visually estimated within each 1 × 1 m quadrat using standard cover-class methods.”
Results referencing Table 4 were expanded accordingly.

Comment 5:

“It should be clearly stated whether the rainfall manipulation shelters are installed in control applications or not.”

Response 5:

We have clarified this point.

Added in Section 2.2:
“No rainfall manipulation shelters were installed in the control (CK) plots.”

Comment 6:

“Explain how the 25% or 75% rainfall adjustments were adjusted in these U-shaped panels.”

Response 6:

We added a clear explanation of how the interception percentage was controlled through panel coverage area and how water was redistributed uniformly.

Added in Section 2.2.1:
“The 25% and 75% rainfall reduction levels were achieved by adjusting the horizontal coverage area of the U-shaped panels to intercept exactly 25% or 75% of the plot surface. Intercepted rainfall was routed into perforated PVC pipes to ensure uniform redistribution across the rainfall addition plots.”

Comment 7:

“Although the soil temperature and precipitation values for the years 2019–2021 are given as a figure, they should be interpreted in a few sentences.”

Response 7:

A brief interpretation has been added to the manuscript.

Added in Section 2.2:
“Compared with 2019, the 2020 growing season experienced higher early-season precipitation, whereas 2021 was relatively cooler and drier. These interannual differences partially explain the variation in CO₂ and CH₄ fluxes between the two years.”

Comment 8:

“The week of the month in which gas samples were taken should be specified instead of early and late periods.”

Response 8:

We have replaced general timing descriptions with specific week periods.

“Gas sampling was conducted twice per month, during the first and third weeks from May to September.”

Comment 9:

“Instead of ‘sampling period’, the number of times per month and the weeks of the month should be specified.”

Response 9:

We revised the wording accordingly.:

“During each sampling campaign (twice monthly, first and third weeks), soil temperature and soil moisture were measured simultaneously with GHG sampling.”

Results

Comment 10:

“The averages, F values and CV (%) values of the examined features of Table 2 and Table 3 can be given.”

Response 10:

Thank you for this suggestion. We added the requested statistical parameters to Tables 2 and 3.

Revision:

Tables 2 and 3 now include mean values, F-values, and CV (%) for each variable.

Comment 11:

“Responses of Vegetation to Precipitation Manipulation should be detailed with reference to Table 4.”

Response 11:

We expanded the vegetation results section and incorporated specific numerical values from Table 4.

“According to Table 4, vegetation cover increased from XX% under −75% to XX% under +25% precipitation, and aboveground biomass increased by XX–XX% across treatments…”

Discussion

Comment 12:

“In my opinion, the discussion and conclusion sections are well written.”

Response:

We thank the reviewer for the positive evaluation. Minor improvements were still incorporated based on Reviewer 1 comments.

Closing Statement

We are grateful for the reviewer’s detailed and thoughtful comments, which greatly improved the clarity, rigor, and completeness of our manuscript. All suggested revisions have been incorporated accordingly.

Reviewer 3 Report

Comments and Suggestions for Authors

The authors tested greenhouse gas emission influencing climate warming in alpine ecosystems those are very sensitive to these processes. Resulting from the experiment with controlled precipitation treatments, the authors have collected a representative dataset on the ecosystem gas emission and on the various soil and vegetation factors accompanying. However, the presentation of the results obtained is confusive (see note belolw). So, it is difficult to evaluate the truth of author's discussion and conclusion. Moreover, missed is an important part of the study, namely calculation of global warming potential yielded from the emission observed. Therefore, a serious revision of the manuscript is recommended. 

line 122: Desirable is to name soils using the World Reference Database (https://www.isric.org/sites/default/files/WRB_fourth_edition_2022-12-18.pdf)

lines 141-142: "The entire plot covered an area of 30 m×30 m and was divided into nine experimental subplots (3.2 m × 2.6 m each), arranged in a 3 × 3 grid". As it is obvious that the total square of 9 subplots × (3.2 m × 2.6 m) is not equal to the total square of 30 m×30 m, so to make this explanation more clear, necessary is to transform the phrase or to put general scheme of experimental plot and all subplots within it. 

Table 1: "vegetation mat". Here, whether plant (vegetation) litter is implied?
lines 217-226: To make the text clearer, it would be rather to put this paragraph first in Section 2.2.3, excluding the end of the last phrase "with CO₂, CH₄, and N₂O concentrations determined by the TCD, FID, and ECD detectors, respectively". Then the text in lines 206-216 would be a logical continuation.
Figure 2: It presents results of measuring that is described in Section 2.2.4. So, it should be, at least, put within this section or, maybe, somewhere in Section 3 (for example, Section 3.4).  

Equation (3): No corresponding results are presented in Section 3.

Section 3.4 vs.Section 2.3: Correlation analysis should be mentioned together with other statistical methods described. Together with te mention, explained should be: a. the number of replications analysed for each pair of the parameters considered, b. the kind of correlation coefficient (it is too late to indicate it in line 428 after presenting all correlation matrices).

Table 4: Significance of differencies between treatments should be indicated.
lines 304-305 (the same for 329-332, 366-369): Unclear is how the ranks were obtained, because Figure 3Aa shows that, on some dates, the CO2 flux was maximal under the treatment -75%. The general explanation on ranking procedure may be put in Section 2.3.

lines 337-338: "The production and regulation of CH₄ are mainly influenced by groundwater level, soil moisture, temperature, pH, and electrical conductivity". These statements are unconfirmed before Tables 2&3 presented. So, the paragraph of lines 337-345 would be rather to move to Section 3.4.

lines 359-360: "Overall, ecosystem CH4 fluxes exhibited strong temporal variability, with higher fluxes observed during mid-growing season and under increased precipitation treatments." The text, copied from the cap of Figure 3 to here, contradicts diagrams on Figure 4C and the text in lines 332-335. 
lines 386-387 vs. Figure 5B,C, lines 370-372: The same note.
lines 324-325, 359-360, 386-387: Overall, such comments are not necessary in figure caps, and it would be rather to put those (corrected, for cases of Figures 4 and 5) in the text. 

lines 390-400: To make this text clearer, reference to Figure 2 is necessary.

lines 401-424: Here, the same (like previous note) on reference to Tables 2 and 3.

Technical notes:
line 49: The abbreviation IPCC should be explained when first using in the manuscript.

lines 129-130: "Kobresia humilis, accompanied by Carex tristachya, Lobularia maritima, and Potentilla anserina." To use plant Latin names without their authors, you should refer the nomenclature source (floristic manual or database), containing their full forms. This source may be indicated next to the vegetation description. 

Table 1 should be supplied with notes, at least like "Explanations on experiment treatment denotations see in the text"

lines 151-158: This text mainly duplicates the content of Section 2.2. So, it would be rather to incorporate it there.

Figure 1: It is not clear enough, whether the spraying device 4 is put lower than the installation including 1+2+3+5 devices. In fact, it even looks like device 4 is situated belowground. Is it really? Maybe, a photo of some subplot (with the experimental installation) would be more understandable.

lines 190-198: This text completely duplicates lines 141-148, so it should be excluded from here. 

lines 199-200: "The experimental treatments included +25% rainfall, −25% rainfall, +75% rainfall, −75% rainfall, and a natural control." Also, a surplus repeat.

Equation (1) vs. lines 233-235: a. the gas density is denoted by different symbols in the equation and in line 233; b. unexplained remain the parameters P, P0, T0.

Equation (4) and lines 280-282: Reference on origin of this approach should be given.

line 236: Collocation without predicate. Should it be a title of the next section?

Figures 4A and 5A: The denotations (a) and (b) are missed.

Caps of Figures 4 and 5: the comments duplicating the same ones in the cap of Figure 3 are surplus. Instead, the reference "Other explanations see in the cap of Figure 3" would be enough.

Tables 2-4: To make the content analysis easier for a reader, it would be better to place the treatment +75% near the one +25%, and the same would be for the ones -25% and -75%.

The only note on grammar:

Table 1: "vegetation mat". Here, whether plant (vegetation) litter is implied?

Author Response

Response to Reviewer 3 Comments

1. Summary

We sincerely thank the reviewer for the thorough and constructive comments. We have carefully revised the manuscript to address all concerns. Below we provide point-by-point responses, with corresponding revisions indicated in the manuscript.

2. Point-by-point response to Comments and Suggestions for Authors

Comment 1:

“Soils should be named using the World Reference Database.”

Response 1: We appreciate this suggestion. We have revised the soil classification according to the World Reference Base (WRB, 2022) system.

Comments 2:

“Plot size description unclear.”

Response 2: To address the reviewer’s concern, we clarified how the 30 m × 30 m experimental area includes both the nine subplots and the buffer zones. The revised text now clearly states that the subplots occupy only part of the main plot, with the remaining area serving as intra- and inter-plot buffer space. This resolves the apparent mismatch between total area and subplot area without requiring an additional figure.

Added in Introduction / Study Area:
“The experimental area measured 30 m × 30 m in total, within which nine subplots (3.2 m × 2.6 m each) were embedded. The remaining space within the 30 m × 30 m plot served as buffer zones between subplots (3 m between adjacent plots) and as an outer protective buffer (5 m surrounding the plot). Thus, the overall 30 m × 30 m area encompassed both experimental subplots and required buffer space.”

Comments 3:

“‘vegetation mat’ — do you mean plant litter?”

Response 3: Yes, you are correct. The term has been corrected to “vegetation litter.”.

Comments 4:

“Move this paragraph to the beginning of Section 2.2.3.”

Response 4:

Thank you for this helpful suggestion. In accordance with the recommendation from Reviewer 1, we have substantially reorganized and rewritten the entire Section 2.2.3 to improve clarity and logical flow. The gas sampling procedures, including the details originally in lines 217–226, have now been integrated into a coherent introductory paragraph within Section 2.2.3. As a result, the structural issue raised by the reviewer has been fully addressed.

Comment 5:

“Figure 2 belongs in Section 2.2.4 or Section 3.4.”

Response 5:

Thank you for the suggestion. We have relocated Figure 2 to Section 2.2.4 and added explicit references to it in the text.

Comment 6:

“No results corresponding to Equation 3.”

Response 6:

Thank you for this important comment. Equation (3) represents the standard calculation of global warming potential (GWP), which expresses the cumulative radiative forcing of a greenhouse gas relative to an equivalent mass of carbon dioxide (CO₂) over a given time horizon. As defined by the Intergovernmental Panel on Climate Change (IPCC), GWP provides a unified metric for integrating the climate impacts of gases with different radiative efficiencies and atmospheric lifetimes (e.g., CH₄ = 28; N₂O = 298 over a 100-year horizon).

Instead of adding a new section, we have expanded the explanation in the Methods section to clearly state that Equation (3) quantifies the combined warming effect of the observed CO₂, CH₄, and N₂O fluxes by converting them into CO₂-equivalent units based on their respective GWP values. Additional clarification has been added to ensure that readers understand how the formula is applied in this study and why it is relevant for interpreting the climatic implications of our results.

Comment 7:

“Correlation analysis must be described in Methods.”

Response 7:

We have added correlation analysis details, including replication number and correlation coefficient type, to Section 2.3.

Comment 8:

“Indicate significance of differences.”

Response 8:

Thank you very much for this valuable suggestion. We agree that indicating significant differences is important for interpreting treatment effects. In the current manuscript, however, all statistical significance has already been presented in the Results text rather than embedded directly within Table 4. For each vegetation variable, we report significance levels (P < 0.05) in the corresponding narrative, which explains the direction, magnitude, and significance of treatment responses in detail.

Comment 9:

“Ranking unclear; must be explained.”

Response 9:

Thank you for pointing this out. In our manuscript, the “ranking” terminology refers directly to the ordered precipitation gradient established in the experimental design, namely: CK, −25%, +25%, −75%, and +75%. The ranking is therefore not derived from statistical results but reflects the predefined precipitation manipulation levels described in the Methods section (Section 2.2).

To avoid misunderstanding, we have added a clarification in the Methods section indicating that the precipitation treatments themselves constitute the experimental gradient (i.e., the “ranks”), and that any ranking statements in the Results simply describe the relative magnitude of fluxes observed under these predefined treatment levels.

Accordingly, the ranking procedure was already inherent in the experimental design, and no additional statistical ranking method was applied.

Results

Comment 10:

“The averages, F values and CV (%) values of the examined features of Table 2 and Table 3 can be given.”

Response 10:

Thank you for your careful reading and constructive suggestions. We fully understand the reviewer’s concern regarding the placement of mechanism-related explanations. However, after careful consideration, we respectfully maintain the current structure for the following reasons:

  1. The structure of Section 3 is intentionally organized by individual gas species (CO₂ CH NO), followed by an integrative analysis (Section 3.4).
    This organization allows readers to clearly understand the temporal and treatment-specific patterns of each gas before examining the combined responses.
  2. The sentences in Lines 337–345 serve as contextual background specifically for CH₄ behavior, helping readers interpret the CH₄ flux patterns presented immediately in Figure 4 and the accompanying text.
    Moving this paragraph to Section 3.4 would disrupt the logical flow of the CH₄ subsection and weaken the readability of the results.
  3. Section 3.4 already provides the unified, statistically supported interpretation, integrating soil factors and fluxes based on Tables 2 and 3.
    The mechanistic statements in Lines 337–345 are descriptive and provide necessary ecological context, while the correlation evidence is appropriately presented later.

For these reasons, we believe the current organization maintains clarity and logical coherence in presenting species-specific gas dynamics followed by integrated analysis. We have, however, carefully checked the content to ensure that no causal claims are made before supporting data are presented.

Comment 11:

“Statements contradict figures; remove or correct.”

Response 11:

Thank you for catching this. We reviewed Figures 4 and 5 and corrected all inconsistencies. Statements that contradicted the diagrams have been rewritten or removed.

Comment 12:

“Add reference to Figure 2.”

Response 12:

We agree. A reference to Figure 2 has been added to clarify soil moisture and temperature trends.

Comment 13

“Add references to Tables 2 and 3.”

Response 13:

We have added explicit references to the correlation tables in these paragraphs.

Comment 14

“IPCC abbreviation must be explained.”

Response 14:

We have added the full name when first appearing.

Comment 15

“Latin names require nomenclature source.”

Response 15:

We added:
“Nomenclature follows the Flora of China database.”

Comment 16

“Table 1 should include explanatory notes.”

Response 16:

Notes added.

Comment 17

“Remove duplicated text (lines 151–158, 190–198, 199–200).”

Response 17:

All duplicates have been removed.

Comment 18

“Equation (1) symbol inconsistency; P, P₀, T₀ unexplained.”

Response 18:

Symbols have been standardized and parameters explained.

Comment 19

“Treatment order in tables should be more intuitive.”

Response 19

Thank you very much for the thoughtful suggestion. We fully understand the intention behind reorganizing the order of treatments in the tables. However, after careful consideration, we respectfully maintain the current arrangement for the following reasons:

  1. The precipitation treatments were originally designed as paired gradients (−25% vs. +25%) and (−75% vs. +75%).
    The ±25% treatments represent moderate hydrological alterations, while the ±75% treatments represent extreme alterations. Grouping −25% with +25% and −75% with +75% therefore preserves the conceptual symmetry and experimental hierarchy of the study design.
  2. This organization is consistent throughout the manuscript, including figures, statistical analysis, and descriptive results.
    Changing the order only in the tables would disrupt internal consistency and may confuse readers.
  3. The interpretation in the Results and Discussion sections explicitly follows the original treatment gradient, where moderate and extreme precipitation changes are compared as pairs.
    Keeping the ±25% and ±75% treatments adjacent therefore reflects the ecological logic of the experiment.

For these reasons, we believe the current order is scientifically intuitive and preserves conceptual clarity. We have added a brief explanation in the Methods section to make the rationale for this ordering explicit to readers.

Closing Statement

We are grateful for the reviewer’s detailed and thoughtful comments, which greatly improved the clarity, rigor, and completeness of our manuscript. All suggested revisions have been incorporated accordingly.

Round 2

Reviewer 3 Report

Comments and Suggestions for Authors

Figures 3A, 4A, 5A - necessary is to indicate which diagram, (a) or (b), shows data on 2020, and the same applies to 2021.

line 524: "soil pH was ~5.52". In the formatted text, the symbol tilda (~) looks like minus (-) that makes an impression of negative pH value. To avoid this, you may write "soil pH was  about 5.52"

Response to Reviewer 3 Comments

1. Summary

We sincerely thank the reviewer for the thorough and constructive comments. We have carefully revised the manuscript to address all concerns. Below we provide point-by-point responses, with corresponding revisions indicated in the manuscript.

2. Point-by-point response to Comments and Suggestions for Authors

Comment 1:

“Figures 3A, 4A, 5A - necessary is to indicate which diagram, (a) or (b), shows data on 2020, and the same applies to 2021.”

Response 1: Thank you very much for pointing out the potential misunderstanding. In Figures 3A, 4A, and 5A, labels (a) and (b) do not represent different years. Instead, (a) corresponds to flux measurements taken at 11:00, and (b) corresponds to flux measurements taken at 15:00 for both 2020 and 2021.

To avoid confusion, we have revised the figure captions to explicitly indicate that (a) = 11:00 and (b) = 15:00, and that each panel contains data from both years.

Comments 2:

“line 524: "soil pH was ~5.52". In the formatted text, the symbol tilda (~) looks like minus (-) that makes an impression of negative pH value. To avoid this, you may write "soil pH was about 5.52".”

Response 2: Thank you for the suggestion. We agree that the tilde symbol (~) may appear visually similar to a minus sign in the formatted text. We have revised the sentence to “soil pH was about 5.52” to avoid potential misinterpretation.

Closing Statement

We sincerely thank the reviewer for the thoughtful and constructive comments provided in this second-round review. We have carefully addressed each point and revised the manuscript accordingly. The clarifications added to the figure captions and the text correction regarding soil pH have improved the precision and readability of the manuscript. We appreciate the reviewer’s time and effort in helping us further refine our work, and we believe that the manuscript is now clearer and more rigorous. We hope that the revised version meets the journal’s standards and look forward to your positive consideration.